# PARALLEL TOKEN PREDICTION FOR LANGUAGE MODELS

**Felix Draxler**[*]    **Justus Will**[*]    **Farrin Marouf Sofian**
Department of Computer Science,
University of California, Irvine

**Theofanis Karaletsos**
Chan-Zuckerberg Initiative
& Pyramidal AI

**Sameer Singh**
Department of Computer Science,
University of California, Irvine

**Stephan Mandt**
Computer Science & AI in Science Institute,
University of California, Irvine

## ABSTRACT

Autoregressive decoding in language models is inherently slow, generating only one token per forward pass. We propose Parallel Token Prediction (PTP), a general-purpose framework for predicting multiple tokens in a single model call. PTP moves the source of randomness from post-hoc sampling to random input variables, making future tokens deterministic functions of those inputs and thus jointly predictable in a single forward pass. We prove that a single PTP call can represent arbitrary dependencies between tokens. PTP is trained by distilling an existing model or through inverse autoregressive training without a teacher. Experimentally, PTP achieves a 2.4× speedup on a diverse-task speculative decoding benchmark. We provide code and checkpoints at `https://github.com/mandt-lab/ptp`.

## 1 INTRODUCTION

Autoregressive transformers (Vaswani et al., 2017) are the foundation of today's large language models (LLMs) (Brown et al., 2020). Their sequential generation process, however, remains a major bottleneck: predicting one token requires one forward pass of the model. This increases inference latency significantly compared to what a single transformer call would achieve.

Many recent efforts aim to bypass this bottleneck by predicting multiple tokens at once. Broadly, they can be categorized into two lines of work: the first, speculative decoding, takes a systems approach, making predictions in a lightweight model that is verified by a large model (Leviathan et al., 2023; Chen et al., 2023; Sun et al., 2023; Zhong et al., 2025). The second line of work makes use of predicting several tokens independently of each other. This significantly reduces the search space for sequences and improves overall model quality (Qi et al., 2020; Gloeckle et al., 2024;

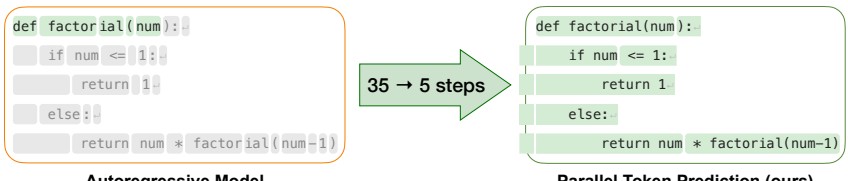

Figure 1: **Our parallel model generates the same text as its teacher in a fraction of the steps.** By the time our model *(right)* has generated an entire function, its autoregressive teacher *(left)* is still busy with the function header. Green tokens are generated during the time spent on PTP; gray tokens show the remaining steps an autoregressive model would still need. Prompt: `Write a Python function that computes the factorial of a number.`

---

[*]Equal contribution. Corresponding authors: {fdraxler,jcwill,mandt}@uci.edu

DeepSeek-AI et al., 2025). Similarly, discrete diffusion can iteratively refine generated sequences across several steps (Hoogeboom et al., 2021; Austin et al., 2021). However, all of these methods still contain an irreducible sequential component to generate sequences.

Our work takes a step towards filling this gap. We propose a framework that, in theory, can generate arbitrary length sequences in parallel. This is enabled by a small but fundamental architectural change: instead of sampling from the distributions predicted by an autoregressive model in a post-processing step, we feed the involved random variables as an input to the model: the model learns to sample. This enables it to anticipate which tokens will be sampled and to predict them jointly. Figure 1 shows how this results in a significant reduction in model calls to produce identical text. Similar frameworks have been formulated in the normalizing flow literature: Inverse Autoregressive Flows (Kingma et al., 2016) generate samples of many continuous dimensions in parallel and Free-form Flows (Draxler et al., 2024) distill a fast generator network. We transfer these concepts to sampling discrete sequences from a continuous latent space.

Our contributions are as follows:

- We propose *Parallel Token Prediction* (PTP), a modeling approach for discrete data that generates multiple interdependent tokens in one model call (Section 2.1).

- We prove that PTP is as expressive as autoregressive models (Theorems 1 and 2).

- We propose *Partial Quadratic Decoding*, an efficient error correction scheme that allows parallel verification of long generated sequences during sampling (Section 3).

- Experimentally, we distill real-world natural language PTP models, achieving a speedup of $2.4\times$ and 4.2 accepted tokens per speculative decoding step (Section 4).

Together, our framework opens a design space to accurately predict several tokens in parallel, reducing latency in language model output without limiting representational power.

## 2 PARALLEL TOKEN PREDICTION

### 2.1 PARALLEL SAMPLING

To construct our *Parallel Token Prediction* framework, let us recap how a classical transformer decoder generates text, see Fig. 2 (left). This decoder iteratively predicts the categorical distribution of the next token $t_i \in \{1, \ldots, V\}$ based on all previous tokens $t_{<i} = (t_1, \ldots, t_{i-1})$,

$$P_i := P(t_i | t_{<i}). \tag{1}$$

One then samples $t_i \sim P_i$, before moving on to predicting the next token. The requirement to know token $t_i$ to predict the next token $t_{i+1}$ is the reason for the latency of autoregressive inference. For simplicity, we assume this distribution is the final distribution that is used to generate tokens, in that it already reflects temperature scaling (Guo et al., 2017), top-k and top-p sampling (Holtzman et al., 2020), or other approaches trading sample diversity with quality.

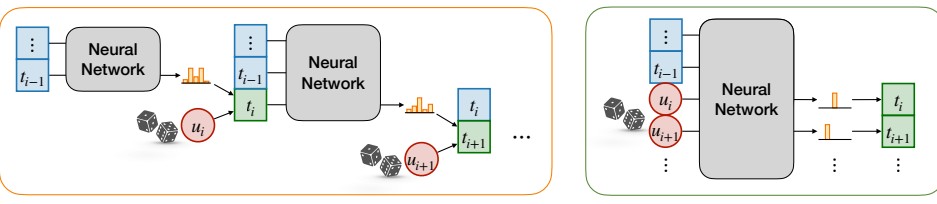

**Autoregressive Model**          **One-Hot Parallel Token Prediction**

Figure 2: **Parallel Token Prediction predicts several tokens in one model call.** *(Left)* An autoregressive model predicts a single $t_i$ per step. It predicts the distribution of each single next token, then the token is chosen with the help of an auxiliary random variable $u_i \sim \mathcal{U}[0,1]$ (see Fig. 3). *(Right)* One-Hot Parallel Token Prediction learns the sampling process by feeding the auxiliaries into the model. This makes predictions deterministic, allowing joint prediction of several tokens.

To break the autoregressive nature, we need to understand how exactly tokens are sampled from $P_i$: One draws an auxiliary random variable $u_i \sim \mathcal{U}[0,1]$ and looks up the corresponding token from the inverse cumulative distribution function as follows:

$$t_i = \text{Pick}(u_i, P_i) \equiv \min_{j \in \{1, \dots V\}} \{j : F_{ij} > u_i\}, \quad \text{where } F_{ij} = \sum_{l=1}^{j} P_{il}. \tag{2}$$

Figure 3 illustrates how this function jumps from token to token as a function of $u_i$. Here, $j$ iterates possible token choices, $P_{il}$ is the probability to sample $t_i = l$, and $F_{ij}$ is the cumulative distribution to sample a token $t_i \in \{1, \dots j\}$.

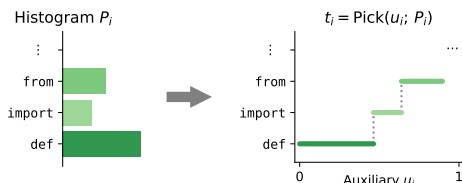

Note that while Eq. (1) defines a distribution over possible next tokens, Eq. (2) is a deterministic rule once the auxiliary variable $u_i$ is drawn. Thus, write this rule as an explicit deterministic function:

$$t_i = f_P(t_{<i}; u_i) = \text{Pick}(u_i, P(\cdot \mid t_{<i})). \tag{3}$$

These auxiliary variables are all we need to perform parallel generation of text: all information about which token $t_i$ we are going to select is available to the model if it has access to $u_i$ as one of its inputs. Since $u_i$ uniquely determines $t_i$ given $t_{<i}$, it carries the same information as $t_i$ itself. The model

**Figure 3: Sampling from a discrete distribution.** Given a histogram $P_i$ *(left)*, compute the inverse cumulative distribution function *(right)* and look up the token at a random location $u_i \in \mathcal{U}[0,1]$. PTP learns the function on the right instead of the histogram.

can therefore use $u_i$ as a stand-in for $t_i$ when predicting $t_{i+1}$, then $u_{i+1}$ as a stand-in for $t_{i+1}$, and so on. Feeding all auxiliary variables $u_i, \dots, u_k$ into the model at once thus allows every future token to be predicted without any sequential dependency (formal proof in Appendix A.1):

**Theorem 1.** *Let $P$ denote a probability distribution for next token prediction. Then, the future token $t_k$ can be selected as a deterministic function $f_P$ of previous tokens $t_{<i}$ and auxiliary variables $u_i, \dots, u_k \sim \mathcal{U}[0,1]$:*

$$t_k = f_P(t_{<i}; u_i, \dots u_k), \qquad \text{for all } k \geq i. \tag{4}$$

Theorem 1 shows a clear path to build a model that can sample many tokens in parallel: instead of learning the distribution $P(t_k \mid t_{<k})$, we propose to directly fit the function $f_P(t_{<i}; u_i, \dots, u_k)$, which jointly predicts future tokens $t_k$.

Figure 2 (right) visualizes how this path can be implemented with a standard transformer backbone (Vaswani et al., 2017): alongside the previous tokens, simply feed random auxiliary variables $u_i, \dots, u_N \sim \mathcal{U}[0,1]$ into the model. Then predict $P(t_k \mid t_{<i}; u_i, \dots, u_k)$ for each future token in one model call. Per Theorem 1, since $u_k$ uniquely determines $t_k$, this prediction narrows down to a single token — a one-hot distribution: $P(t_k \mid t_{<i}; u_i, \dots, u_k) = \mathbf{1}(t_k = f_P(t_{<i}; u_i, \dots, u_k))$. In practice, we take the $\text{argmax}$ to extract that predicted token:

$$t_k = f_P^{\text{O-PTP}}(t_{<i}; u_i, \dots, u_k) = \text{argmax}(P(t_k \mid t_{<i}; u_i, \dots, u_k)) \tag{5}$$

We refer to this model as a **One-Hot Parallel Token Prediction Model** (O-PTP). O-PTPs can be trained to replicate an existing autoregressive model $P$, as we discuss in Section 2.2.1.

O-PTPs converge to one-hot distributions, and therefore do not expose the underlying sampling distributions of the model. While we gain access to the model's confidence, this prevents access to the original conditional probabilities $P(t_k \mid t_{<k})$, which are required for training without a teacher, adjusting temperature (Holtzman et al., 2020), and uncertainty quantification. To address this, we introduce **Categorical Parallel Token Prediction** (C-PTP), which recovers the full conditional distribution of each token. The key idea is to predict each token $t_k$ while conditioning on all *past* auxiliary variables $u_i, \dots, u_{k-1}$, but explicitly excluding its own auxiliary variable $u_k$. By Theorem 1, these past auxiliaries deterministically encode the sampled history $t_{<k}$. Withholding $u_k$ preserves the uncertainty over $t_k$ rather than collapsing it to a point mass. As a result, conditioning $t_k$ on $(t_{<i}, u_i, \dots, u_{k-1})$ exactly recovers the original autoregressive conditional $P(t_k \mid t_{<k})$ (proof in Appendix A.2):

**Theorem 2.** *Let $P$ denote a probability distribution for next token prediction. Then, the distribution of a token $t_k$ is fully determined by context tokens $t_{<i}$ and the past auxiliary variables $u_i, \dots, u_{k-1}$:*

$$P(t_k \mid t_{<i}, u_i, \dots, u_{k-1}) = P(t_k \mid t_{<k}), \qquad \text{for all } k \geq i. \tag{6}$$

Note how Eq. (6) does not depend on $u_k$. Figure 4 shows how Theorem 2 can be used to predict the distribution of the tokens $t_i, \ldots t_N$ in parallel. Just like for O-PTP, first sample all required auxiliary variables $u_i, \ldots u_N \sim \mathcal{U}[0, 1]$, and then predict all $P_k = P(t_k \mid t_{<i}, u_i, \ldots, u_{k-1})$ in parallel. Sampling from these distributions is done just as in Eq. (3):

$$
\begin{aligned}
t_k &= f_P^{\text{C-PTP}}(t_{<i}, u_i, \ldots, u_k) \\
&= \text{Pick}(u_k; P(t_k \mid t_{<i}, u_i, \ldots, u_{k-1})).
\end{aligned}
\tag{7}
$$

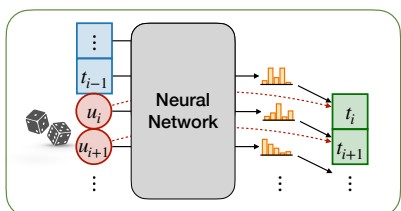

**Categorical Parallel Token Prediction**

By using a causal decoder architecture, we can properly mask which token has access to which auxiliaries.

C-PTP can be trained without a teacher by iteratively solving Eq. (6) for the auxiliary $u_k$ corresponding to a given token $t_k$, see Section 2.2.2. Like O-PTP, it can also be distilled from a teacher.

Both One-Hot and Categorical Parallel Token Prediction allow coordinated token prediction in a single model call. **By Theorems 1 and 2, there are no fundamental restrictions as to how many tokens can be jointly modeled** apart from model capacity. In the next section, we propose approaches for training from scratch (only C-PTP) and distilling an existing model (both O-PTP and C-PTP).

Figure 4: **Categorical Parallel Token Prediction.** By withholding $u_k$ for the prediction of $t_k$, C-PTP learns the same distribution as an autoregressive model (Theorem 2). The auxiliary $u_k$ is used to sample token $t_k$.

## 2.2 TRAINING

Before deriving the training paradigms for Parallel Token Prediction Models, let us quickly recall that autoregressive models are trained by minimizing the cross-entropy between samples from the training data $t \sim P(t)$ and the model $P_\theta$:

$$
\mathcal{L}(\theta) = \mathbb{E}_{t \sim P(t)} \left[ -\sum_{i=1}^{T} \log P_\theta(t_i \mid t_{<i}) \right].
\tag{8}
$$

Using a causal model such as a transformer (Vaswani et al., 2017), this loss can be evaluated on an entire sequence of tokens in a single model call (Radford et al., 2018). We now present how to distill both One-Hot and Categorical Parallel Token Prediction Models from a trained autoregressive model. We then show how the categorical variant can be self-distilled from data alone via Eq. (8).

### 2.2.1 DISTILLATION

To build a PTP model $P_\theta$ that speeds up inference of a pretrained autoregressive model $Q_\phi$, we propose to train it using distillation. Given enough model capacity, this results in a fast model that generates identical text as the base model (Theorems 1 and 2). We defer efficiently correcting errors arising from finite resources to the subsequent Section 3.

To train the student for a given training sequence $t = t_1, \ldots, t_T$, we reverse engineer the auxiliary variables $u_1, \ldots, u_T$ under which the teacher would have generated the sequence. We then randomly split the sequence into context and prediction sequences, and evaluate a loss that leads the student towards the correct generation. This process is summarized in Algorithm 3 in Appendix G.1.

**Auxiliary variables.** We train the model based on auxiliary variables that the teacher model would use to generate the training sequence. To this end, we evaluate the teacher distributions of each training token to get the cumulative discrete distributions $F_1, \ldots, F_T$ for each token. Inverting Eq. (2) for $u_k$, we find for every token in the sequence ($k = 1, \ldots, T$):

$$
u_k \in [F_{k, t_k - 1}, F_{k, t_k}) = \text{Pick}^{-1}(\{t_k\}, P_k).
\tag{9}
$$

Since $u_k$ is continuous, while $t_k$ is discrete, we can randomly pick any compatible value. See Appendix E.3 for details.

**Loss evaluation.** Both O-PTP and C-PTP can be trained with standard losses given an input sequence $t$, with $u_k$ extracted using Eq. (9). We randomly split the sequence at an index $i$ into context

and prediction, and evaluate the following cross-entropy losses, choosing $N$ the index of the last token in the completion part:

$$\text{O-PTP:} \quad \mathcal{L}(\theta; t, i) = -\sum_{k=i}^{N} \log P_\theta(t_k \mid t_{<i}, u_i, \ldots, u_{k-1}, u_k), \tag{10}$$

$$\text{C-PTP:} \quad \mathcal{L}(\theta; t, i) = -\sum_{k=i}^{N} \log P_\theta(t_k \mid t_{<i}, u_i, \ldots, u_{k-1}). \tag{11}$$

C-PTP can also be trained similar to knowledge distillation (Hinton et al., 2015): to this end, we explicitly match the student's $P_{\theta,k}$ to the teacher distribution $Q_{\varphi,k}$. This works for any loss $d(Q, P)$ for the difference between categorical distributions, such as the Kullback–Leibler divergence $d = \text{KL}(Q \,\|\, P)$ or its reverse variant $d = \text{KL}(P \,\|\, Q)$:

$$\mathcal{L}(\theta; t, i) = \sum_{k=i}^{N} d(Q_\varphi(t_k \mid t_{<k}), P_\theta(t_k \mid t_{<i}, u_{i\ldots k-1})). \tag{12}$$

At their corresponding minima, the losses in Eqs. (10) to (12) are minimized by the optimal solutions given in Theorems 1 and 2: a one-hot distribution for O-PTP and the teacher distribution for C-PTP.

**Training data.** We can train the student using sequences from any data source, even if the teacher assigns different probabilities to them. As long as the teacher assigns non-zero probability and the dataset is sufficiently large, querying the teacher for the corresponding auxiliary variables provides full supervision and allows the student to match the teacher everywhere. We conceptually and empirically compare several training data sources (teacher samples, dataset sequences, student self-samples) in Appendix E.2. We find the lowest-variance option is to sample training sequences from the teacher.

We now show how C-PTP can be trained on training data alone, without a teacher model available.

### 2.2.2 TRAINING FROM SCRATCH

Categorical Parallel Token Prediction Models can also be trained directly via Eq. (8), avoiding the need to have a teacher model as target. For a given training sequence $t_1, \ldots, t_T$, we again split it into the context $t_{<i}$ and the following prediction $t_{\geq i}$.

Exactly as for distillation, we have to find auxiliary variables that are compatible with every $t_k, k \geq i$. We can do this by selecting, randomly, any $u_k \in [F_{k,t_k-1}, F_{k,t_k})$, equivalently to Eq. (9), where $F_k, t_k$ now is the cumulative probability under $P_\theta$ (instead of the teacher model) to choose $t_k$ when predicting that token. As this probability depends on the previous auxiliary variables $u_i, \ldots, u_{k-1}$, we select them iteratively. Specifically, we can alternate between computing the logits of $P_\theta(t_k \mid t_{<i}, u_i, \ldots, u_{k-1})$, and drawing $u_k$ using equation 9.

Finally, we can train our model using the cross-entropy loss

$$\mathcal{L}(\theta) = \mathbb{E}_{t \sim P(t), i \sim P(i|t)} \left[ -\sum_{k=i}^{N} \log P_\theta(t_k \mid t_{<i}, u_i, \ldots, u_{k-1}) \right]. \tag{13}$$

Algorithm 4 in Appendix G.1 summarizes the procedure. A similar approach of iteratively determining latent variables (our auxiliaries) was proposed by Inverse Autoregressive Flows (Kingma et al., 2016), although they considered continuous variables in an invertible neural network.

## 3 ERROR CORRECTION

Theorems 1 and 2 say that O-PTP and C-PTP can in principle predict coherent sequences of arbitrary length in one model call. Practically, finite model capacity limits the length at which a single transformer pass can produce coordinated text. In this section, we propose how to generate identical text to a base model (of arbitrary architecture and size) at minimum latency.

One successful strategy comes in the form of *Speculative Decoding* (Leviathan et al., 2023). A PTP model proposes several tokens in one model call. A subsequent call to the base model then verifies them. A proposed token $t_k$ is considered correct if it matches $\tilde{t}_k = \text{Pick}(u_k, Q_\varphi(t_k \mid t_{<k}))$,

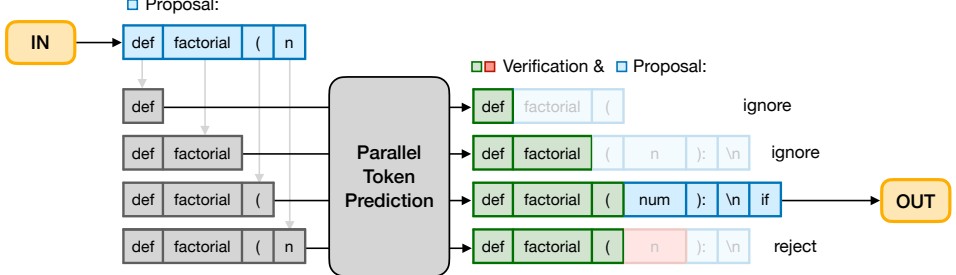

Figure 5: **Partial Quadratic Decoding performs parallel verification and proposing.** Given a proposal of $N_{\text{prop}}$ tokens, the model evaluates multiple branches in parallel, each assuming a different number of proposed tokens are correct. Green tokens are verified, red tokens mark the first mismatch, and blue tokens are new PTP proposals. Compute is allocated based on proposal confidence, and the branch with the longest verified prefix is selected.

the token the base model selects when the same auxiliary $u_k$ is applied to its own distribution. Additionally, since the base model already evaluates its distribution $Q_\varphi(\cdot \mid t_{<k})$ at every position during verification, one token can be sampled from it for free at the first rejection, resulting in #accepted = #correct + 1 tokens (see Appendix C for formal definitions). We will find in Section 4.2.1 that PTP models outperform autoregressive models of the same size in terms of wall-clock speedup in vanilla speculative decoding since they only call the student model once.

Speculative decoding executes proposal and base models in sequence. To reduce this latency, we execute verification and drafting in parallel using *Quadratic Decoding* (Samragh et al., 2025). Since verification has not yet completed when we begin the next drafting step, we do not know how many of the $N_{\text{prop}}$ previously proposed tokens are correct. Rather than waiting, we prepare for all possible outcomes by constructing a branch for each $m \in \{0, \dots, N_{\text{prop}}\}$, each assuming exactly $m$ proposed tokens are correct and predicting continuation tokens conditioned on that assumption. All branches are computed concurrently with verification. Once the base model determines the true number $m^*$ of correct tokens, all other branches are discarded and generation continues from the precomputed continuation of branch $m^*$. Parallelization can be achieved with separate hardware or in one merged call by leveraging *Gated Low-Rank Adaption* (Samragh et al., 2025) to let different model weights attend to different positions of the input.

While quadratic decoding works well for short proposal lengths, its quadratic cost in the number of predicted tokens becomes prohibitively expensive in our case, where we can predict a larger number of coherent tokens. We therefore propose **Partial Quadratic Decoding**, leveraging confidence estimates from the proposal model to significantly reduce the number of proposed tokens while retaining a high number of accepted tokens per step. Figure 5 illustrates our proposed decoding scheme, details can be found in Algorithm 2 and Appendix F.

The confidence estimates for Partial Quadratic Decoding come directly from the O-PTP model: for each proposed token $t_k$, the model's output probability serves as a confidence score $c_k$, which closely tracks the base model's acceptance rate (see Fig. 9 in Appendix F).

Given context $t_{<i}$ and proposed tokens $t_k$ for $k \geq i$, with confidences $c_k$ for $k \geq i$, we treat errors as independent and estimate branch probabilities as

$$P(\#\text{correct} = m \mid t) \approx (1 - c_{i+m}) \prod_{k=i}^{i+m-1} c_k \quad \text{for } m < N_{\text{prop}}. \tag{14}$$

Given a budget of $C$ proposed tokens across all branches, we allocate $B_m$ continuation tokens to branch $m$ by maximizing the expected utility

$$\max_{B \in \{0, \dots, N_{\text{prop}}\}^{N_{\text{prop}}+1}, \sum_m B_m = C} \sum_m P(\#\text{correct} = m \mid t) \, H(B_m), \tag{15}$$

where $H(n)$ is a monotonically increasing reward for predicting $n$ tokens. We solve Eq. (15) greedily, repeatedly assigning the next token to the branch with the highest marginal gain $\Delta_m :=$

$P(\#\text{correct} = m \mid t_{\leq k})[H(B_m + 1) - H(B_m)]$. We find $H(n) = \sum_{j=1}^{n} j\, P(\#\text{correct} = j)$ works well empirically, with $P(\#\text{correct})$ estimated over the training dataset. When additional hardware allows sampling multiple independent proposal sequences in parallel, the acceptance rate can be improved further; see Appendix E.4.

## 4 EXPERIMENTS

We empirically evaluate Parallel Token Prediction (PTP) by its wall-clock speedup compared to an autoregressive model. Since this is hardware-specific, we also report the number of accepted tokens per speculative decoding step (#accepted in Section 3), as this is a hardware-independent and method-agnostic metric.

We first show that C-PTP can be trained from data alone without access to a teacher model (Section 4.1). We then distill a 1.1B-parameter model on code generation and demonstrate that O-PTP achieves larger speedups than autoregressive speculative decoding by enabling the draft model to predict multiple tokens per call (Section 4.2.1). On the same task, we show that conditioning on auxiliary variables yields substantially more correct tokens than independently predicting multiple tokens (Section 4.2.2). Finally, on a speculative decoding benchmark spanning diverse language tasks, we finetune a 7B-parameter model and show that O-PTP outperforms competitive baselines, achieving a wall-clock 2.4× speedup (Section 4.3).

**Implementation**   We implement PTP by adding an embedding scheme for the continuous auxiliary tokens $u$ to a standard transformer:

$$\text{embed}(u) = W\, \text{binary}(u) + b, \quad \text{where } \text{binary}(u) \in \{0, 1\}^{32}. \tag{16}$$

Here, $\text{binary}(u)$ maps the `float32` number $u$ into the 32 digits of a binary float (Witten et al., 1987). The linear layer $(W, b)$ maps this binary vector to the embedding space of the autoregressive transformer. For distilling a fast student model from a teacher, we train O-PTP with cross-entropy (Eq. (10)), sample training data from the teacher once and train multiple epochs on it. We identify these choices through an ablation and practical considerations in Appendix E.

### 4.1 INVERSE AUTOREGRESSIVE TRAINING

We first confirm that C-PTP can be trained as a standalone generative model directly from data. We follow the procedure in Section 2.2.2 and train a model on a dataset that predicts sequences of pick-up locations for taxis in New York City (NYC TLC, 2017).

| Model | Perplexity ($\downarrow$) |
|---|---|
| C-PTP (Ours) | 19.88 |
| Autoregressive | 19.81 |

Table 1: **Categorical Parallel Token Prediction (C-PTP) can be successfully trained from data alone.** The perplexity of C-PTP and an autoregressive model on taxi pickup location sequences (NYC TLC, 2017) are almost identical.

Table 1 shows that the resulting model matches the performance of an autoregressive baseline. Based on the evidence for multi-token prediction (Gloeckle et al., 2024; DeepSeek-AI et al., 2025), we expect the quality of PTP models to outperform autoregressive baselines at larger scales. We leave training larger models from scratch to future work.

### 4.2 LIMITATIONS OF COMPETING FRAMEWORKS

Parallel Token Prediction overcomes important limitations in other frameworks to decrease latency: first, small surrogate models for speculative decoding do not leverage their parallel multi-token potential, which we unlock with our framework in Section 4.2.1. Second, modeling tokens independently produces incoherent sequences (Section 4.2.2). This also limits the representational power of discrete diffusion models (Hoogeboom et al., 2021; Austin et al., 2021). We compare to speculative decoding and independent token prediction on CodeContests (Li et al., 2023), a dataset of coding challenges, by distilling `TinyLlama-1.1B-Chat-v1.0` (Zhang et al., 2024).

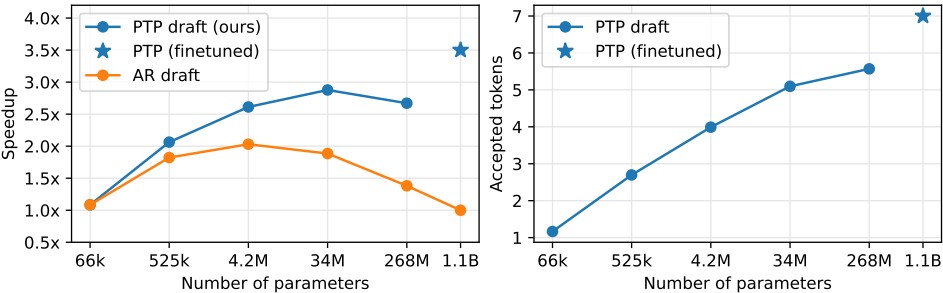

Figure 6: **PTP allows parallelism in draft models for speculative decoding, yielding larger speedups at equal model size.** (Left) Wall-clock speedup relative to standard autoregressive decoding (Right) Average number of tokens accepted per step under teacher verification. Curves compare autoregressive draft models to Parallel Token Prediction (PTP) drafts; (●) are models trained from scratch for a fixed number of epochs, and (⋆) is finetuned from the teacher. Across model sizes, PTP drafts achieve higher speedups by generating more than one correct token per step, whereas autoregressive drafts remain sequential.

### 4.2.1 SMALLER AUTOREGRESSIVE DECODERS

Vanilla speculative decoding distills small autoregressive draft models to imitate the teacher with fewer parameters (Leviathan et al., 2023). Through their reduced size, these models require less compute to predict each token, and the teacher is used to verify them in parallel as in Section 3.

Figure 6 shows that PTP draft models outperform AR draft models because they need fewer student calls for the same number of tokens. This shifts the optimal model size towards larger models. In fact, we find that the best performance is achieved by finetuning the teacher model directly. The latter may be an artifact of the smaller models being trained from scratch instead of being finetuned.

### 4.2.2 INDEPENDENT PREDICTION

Many approaches to predicting multiple tokens in parallel, including multi-token prediction (Qi et al., 2020; Gloeckle et al., 2024) and discrete diffusion models (Hoogeboom et al., 2021; Austin et al., 2021), assume that future tokens are conditionally independent. As a result, later tokens are sampled from marginal distributions that average over incompatible earlier choices, making semantic and syntactic inconsistencies unavoidable even with infinite model capacity. In code generation, this manifests as spurious token combinations such as `def numpy` and `import find`. See Appendix B for a formal derivation.

| Parallelization technique | #accepted (↑) |
|---|---|
| O-PTP (ours) | **7.0 ± 0.1** |
| Independent prediction | 6.2 ± 0.1 |

Table 2: **Speculative decoding with O-PTP accepts more tokens per step than independently predicting tokens.** We distill a draft model once with meaningful auxiliaries that determine token dependence and once with a generic `[MASK]` token. Error is difference between two runs.

To isolate this effect, we compare models trained with informative auxiliary variables $u_i$ (our O-PTP) to models that have multiple independent heads for predicting multiple tokens (Qi et al., 2020; Gloeckle et al., 2024). As shown in Figure 7, O-PTP consistently produces meaningful token combinations by coordinating predictions through the auxiliary variables, while independent prediction frequently produces incompatible pairs. Quantitatively, Table 2 shows that auxiliary variables substantially increase the number of tokens when used in sequential speculative decoding. Figure 1 shows a qualitative sample of our model's predictions.

### 4.3 GENERAL-PURPOSE TEXT GENERATION

In the following experiment, we confirm that we can finetune a large pretrained model using PTP, resulting in a model that predicts several tokens per forward pass.

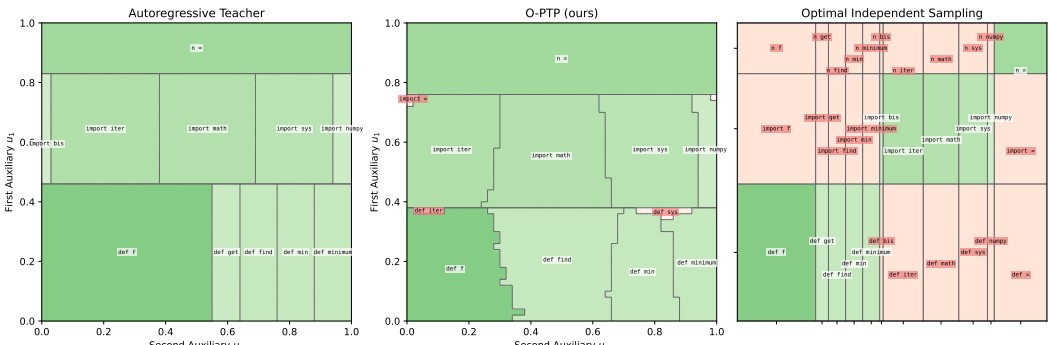

Figure 7: **Parallel Token Prediction generates meaningful pairs of tokens.** Each panel plots the first and second auxiliary variables $(u_1, u_2) \in [0,1]^2$ on the axes; regions in this unit square correspond to the resulting two-token outputs (green: compatible, red: incompatible). *(Left)* In a coding problem, autoregressive sampling generates meaningful token pairs, such as importing a package or defining a method. *(Center)* Our O-PTP coding model produces sensible token pairs, but in a single model call; only rarely ($< 1\%$) does it yield spurious combinations like `def sys`. *(Right)* A model that independently predicts future tokens is bound to fail: in about $60\%$ of cases, it combines incompatible tokens because the second token is not informed about the first.

We distill a One-Hot Parallel Token Prediction model (O-PTP) student from an autoregressive teacher, following the procedure in Section 2.2.1. We use Vicuna-7B (Chiang et al., 2023) as the teacher and finetune the student on ShareGPT conversational data (Chen et al., 2024). Instead of finetuning the full model, we train a gated LoRA (Samragh et al., 2025) adapter, allowing us to make only the minimal changes that are necessary to correctly parse the auxiliary variables.

To reflect a practical deployment scenario for a chat-oriented large language model, we evaluate the performance on SpecBench (Xia et al., 2024), containing a diverse set of tasks: multi-turn conversation (MTC), translation (TL), summarization (SUM), question answering (QA), mathematical reasoning (Math), and retrieval-augmented generation (RAG). Performance is measured using the wall-clock speedup on a single NVIDIA RTX A6000. We also report the hardware-agnostic average number of accepted tokens, as this is highly correlated to the maximum speedup that can be obtained with efficient decoding schemes.

As shown in Table 3, O-PTP consistently predicts more tokens identical to its teacher than existing parallelization baselines across most tasks, resulting in a larger speedup. These results demonstrate that PTP is effective when scaled to large models and heterogeneous real-world datasets. See Appendix D for detailed results, and Appendix H for inference without error correction.

## 5 RELATED WORK

Speeding up the generation of autoregressive models and discrete sequence models in particular has been the focus of a broad body of work, see (Khoshnoodi et al., 2024) for an overview.

Our framework combines two ideas from the Normalizing Flow literature and imports them to modeling discrete data: Inverse Autoregressive Flows (IAF) are trained with fast prediction in mind (Kingma et al., 2016) by iteratively identifying latent variables (our auxiliary variables) that generate a particular continuous one-dimensional value, and Free-Form Flows (FFF) train a single-step generating function as the inverse of an encoder (Draxler et al., 2024).

In the LLM literature, speeding up generation has been approached from various angles. **Speculative decoding** takes a system perspective, using a small draft model to propose multiple tokens and a large target model to verify them (Leviathan et al., 2023; Chen et al., 2023). Variants verify entire sequences (Sun et al., 2023) or use a smaller verifier network Zhong et al. (2025) to improve quality and speed. **Latent variable methods** first sample latent codes from the prompt so that the distribution of subsequent tokens factorizes given latent codes (Gu et al., 2018; Ma et al., 2019).

| Parallelization | MTC | TL | SUM | QA | Math | RAG | Task Average | #accepted (↑) |
|---|---|---|---|---|---|---|---|---|
| O-PTP (ours) | 2.77 | **2.12** | 2.47 | 1.91 | **2.79** | **2.32** | **2.40 ± 0.007** | **4.21 ± 0.011**[†] |
| SAMD (Hu et al., 2025) | **2.79** | 1.87 | 2.47 | 2.05 | 2.60 | 2.22 | 2.34 ± 0.012 | 3.95 ± 0.011 |
| Eagle-2 (Li et al., 2024a) | 2.76 | 1.88 | 2.19 | 2.08 | 2.75 | 2.06 | 2.29 ± 0.010 | 3.93 ± 0.008 |
| Hydra (Ankner et al., 2024) | 2.59 | 1.98 | 1.91 | **2.16** | 2.56 | 1.86 | 2.18 ± 0.007 | 3.40 ± 0.005 |
| Eagle (Li et al., 2024b) | 2.50 | 1.80 | 2.10 | 1.93 | 2.49 | 1.94 | 2.13 ± 0.010 | 3.24 ± 0.006 |
| Recycling (Luo et al., 2025) | 2.08 | 1.77 | 1.93 | 1.83 | 2.21 | 1.73 | 1.93 ± 0.007 | 2.38 ± 0.003 |
| Medusa (Cai et al., 2024) | 2.08 | 1.69 | 1.57 | 1.76 | 2.07 | 1.55 | 1.79 ± 0.004 | 1.93 ± 0.003 |
| PLD (Saxena, 2023) | 1.65 | 1.08 | **2.50** | 1.20 | 1.62 | 1.75 | 1.63 ± 0.004 | 2.70 ± 0.006 |
| SpS (Chen et al., 2023) | 1.57 | 1.12 | 1.54 | 1.39 | 1.40 | 1.55 | 1.43 ± 0.006 | 1.73 ± 0.005 |
| Lookahead (Fu et al., 2024) | 1.46 | 1.13 | 1.26 | 1.24 | 1.55 | 1.17 | 1.30 ± 0.005 | 1.57 ± 0.003 |
| None | 1.00 | 1.00 | 1.00 | 1.00 | 1.00 | 1.00 | 1.00 | 1.00 |

Table 3: **Wall-clock speedups under teacher verification (↑).** We compare our One-Hot Parallel Token Prediction (O-PTP) distilled from Vicuna-7B (Chiang et al., 2023) to concurrent parallelization techniques on the diverse tasks of SpecBench (Xia et al., 2024), see main text for abbreviations. Errors indicate the standard error over three runs. Temperature is set to $0.7$. Across most tasks, O-PTP achieves the largest speed-up. We also report #accepted (↑), the number of accepted tokens per base model call. Here, the errors are standard deviations over all steps across three runs. [†]Parallel verification trades number of accepted tokens for total speedup so we report sequential verification here. For more detailed results see Appendix D.

**Discrete Diffusion Models** leave autoregressive sampling behind by iteratively refining the text starting from a noisy or masked variant (Hoogeboom et al., 2021; Austin et al., 2021). **Multi-head output models** predict several next tokens independent of each other (Qi et al., 2020; Gloeckle et al., 2024; DeepSeek-AI et al., 2025), narrowing down on the possible set of next tokens. Both discrete diffusion and multi-head models assume independence of tokens, which is fundamentally limited in modeling capacity, compare Section 4.2.2 and Feng et al. (2025). Recent work on discrete diffusion models leverages copula models to capture dependencies in an additional model (Liu et al., 2025). We model dependencies between several tokens without an additional model at inference time.

In contrast to the above, our framework is universal in the sense that it can approximate arbitrary dependence between several tokens in a single model call. Our new method is complementary to existing approaches, and we leave exploring these combinations open for future research.

## 6 CONCLUSION

In this paper, we introduce *Parallel Token Prediction*, a framework that permits consistent generation of several tokens in a single autoregressive model call. It eliminates the independence assumptions that limited prior approaches, allowing one to model multiple tokens with arbitrary dependency between them. Empirically, we show that existing models can be distilled into efficient parallel samplers. With error correction, these models produce identical output as a base model while significantly increasing how many tokens are obtained per model call.

This speedup makes language models more practical for real-time applications. Future work includes extending our framework to large scale models, multimodal generation, combining it with complementary acceleration strategies, and exploring theoretical limits on parallelization.

Overall, our results suggest that the sequential bottleneck in autoregressive transformers is not inherent, and that universal, efficient parallel generation is within reach. We think that our experiments in Section 4 only scratch the surface of the possibilities enabled by our Theorems 1 and 2. We envision future work to train large models from scratch that think in long sequences, and speculate that planning over longer spans may improve downstream performance.

**Limitations** The experiments in the present paper focus on distilling existing models, leaving open the question of whether model quality will improve when training from scratch (as is observed for multi-token prediction (Gloeckle et al., 2024)). The inference speedup only manifests given heavily parallelized hardware, where increasing the number of predicted tokens only marginally increases computational cost.

## ACKNOWLEDGMENTS

Justus Will acknowledges support from the HPI Research Center in Machine Learning and Data Science at UC Irvine. Sameer Singh acknowledges funding from the National Science Foundation (NSF) through an NSF CAREER award IIS-2046873. Stephan Mandt acknowledges funding from the National Science Foundation (NSF) through an NSF CAREER Award IIS-2047418, IIS-2007719, the NSF LEAP Center, and the Hasso Plattner Research Center at UCI, and the Chan Zuckerberg Initiative.

## ETHICS STATEMENT

Our work focuses on reducing the inference time of Large Language Models, enabling more computations per unit time and supporting large-scale or real-time applications. While this can improve responsiveness and resource efficiency, it may also increase the potential for misuse, such as generating misinformation or automated spam at higher volumes. Faster inference does not mitigate underlying model biases, so responsible deployment, monitoring, and safeguards are critical to balance performance gains with societal risks.

## REPRODUCIBILITY STATEMENT

We include proofs for all theoretical results introduced in the main text in Appendix A. We include further experimental and implementation details (including model architectures and other hyperparameter choices) in Appendix E and Appendix G. Our code is available at https://github.com/mandt-lab/ptp.

## USAGE OF LARGE LANGUAGE MODELS (LLMS)

We used Large Language Models to improve the writing of this paper, by pasting text and asking for feedback and spell checking; we also used them for code editing, in particular bug resolving. We also used LLMs for discovering related work. All ideas presented in this paper are our own.

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

# A    PROOFS

## A.1    PROOF OF THEOREM 1

*Proof.* We prove by induction over $k, k \geq i$.

For $k = i$, the function is given by Eq. (3).

For $k \mapsto k + 1$, assume the statement holds for $i, \ldots, k$. This gives us access to all $t_1, \ldots, t_{i-1}$ (from the context), and all $t_i, \ldots, t_k$ using $u_i, \ldots, u_k$ (by induction assumption).

Then, we can compute $P_{k+1} = P(t_{k+1} \mid t_{<i}, t_i, \ldots, t_k)$, and pick the next token deterministically via Eq. (2):

$$t_{k+1} = \text{Pick}(u_{k+1}; P_{k+1}) = f_P(t_1, \ldots, t_{i-1}; u_i, \ldots u_k, u_{k+1}). \tag{17}$$

$\square$

## A.2    PROOF OF THEOREM 2

*Proof.* We prove by induction over $k, k \geq i$.

For $k = i$, there is nothing to show, since there are no auxiliaries involved in the statement.

For $k \mapsto k + 1$, assume the statement holds for $k$. This gives us access to the distribution $P_k = P(t_k \mid t_{<k}) = P(t_k \mid t_{<i}, u_i, \ldots u_{k-1})$ of the token $t_k$. Since token $t_k$ is uniquely determined from $P_k$ and $u_k$ via Eq. (3), any distribution conditioning on $P_k, t_k$ can instead condition on $P_k, u_k$ via the law of total probability, so that $P(t_{k+1} \mid t_{<i}, u_i, \ldots u_k) = P(t_{k+1} \mid t_{<k}, t_k)$. $\square$

# B    FORMAL LIMITATIONS OF INDEPENDENT PREDICTION

Without the auxiliary variables, the distribution of all tokens $k > i + 1$ is not informed about the choice we make for tokens $t_{i+1}, \ldots t_{k-1}$. Instead, they are marginalized out, which explains the spurious code snippets:

$$\text{Independent: } P(t_k \mid t_{<i}) = \sum_{t_i, \ldots t_{k-1}} P(t_k \mid t_{<k}) P(t_i, \ldots, t_{k-1} \mid t_{<i}) \neq P(t_k \mid t_{<k}). \tag{18}$$

$$\text{Dependent (ours): } P(t_k \mid t_{<i}, u_i, \ldots u_{k-1}) \overset{\text{thm. 2}}{=} P(t_k \mid t_{<k}). \tag{19}$$

This limits how many tokens can be sampled in parallel without auxiliary variables, even for infinite model capacity. Our Theorems 1 and 2 allow for coordinating tokens, making model capacity the only restriction.

# C    AVERAGE NUMBER OF CORRECT TOKENS

To quantify the effectiveness of parallel token generation, we measure the number of correctly predicted tokens. Given a student output sequence $t_1^{(S)}, t_2^{(S)}, \ldots, t_T^{(S)}$ and a teacher output sequence $t_1^{(T)}, t_2^{(T)}, \ldots, t_T^{(T)}$ of the same length, this metric is defined as

$$\#\text{correct} = \max \left\{ k \mid t_j^{(S)} = t_j^{(T)} \ \forall j \leq k \right\}. \tag{20}$$

That is, $\#\text{correct}$ equals the length of the longest prefix of the student's output that exactly matches the teacher's output. We further define $\#\text{accepted}$ as the number of tokens accepted per base model call. Often, for example with parallel verification, we have $\#\text{correct} = \#\text{accepted}$. Note, however, that with sequential error correction we obtain

$$\#\text{accepted} = \min\{\#\text{correct} + 1, T\}. \tag{21}$$

Intuitively, this measures how many tokens the student can generate in parallel before the first disagreement, and directly corresponds to the number of tokens that can be safely accepted without error correction.

## D    DETAILED COMPARISON ON SPECBENCH

We report more extensive results from our experiment in Section 4. See Table 5 for average number of accepted tokens and Table 4 for wall-clock speed-ups.

| Parallelization | MTC | TL | SUM | QA | Math | RAG | Task-Average |
|---|---|---|---|---|---|---|---|
| O-PTP (ours) | $2.77 \pm 0.021$ | $\mathbf{2.12 \pm 0.027}$ | $2.47 \pm 0.019$ | $1.91 \pm 0.019$ | $\mathbf{2.79 \pm 0.012}$ | $\mathbf{2.32 \pm 0.021}$ | $\mathbf{2.40 \pm 0.007}$ |
| SAMD (Hu et al., 2025) | $\mathbf{2.79 \pm 0.026}$ | $1.87 \pm 0.016$ | $2.47 \pm 0.043$ | $2.05 \pm 0.007$ | $2.60 \pm 0.018$ | $2.22 \pm 0.056$ | $2.34 \pm 0.012$ |
| Eagle-2 (Li et al., 2024a) | $2.76 \pm 0.049$ | $1.88 \pm 0.008$ | $2.19 \pm 0.014$ | $2.08 \pm 0.015$ | $2.75 \pm 0.036$ | $2.06 \pm 0.024$ | $2.29 \pm 0.010$ |
| Hydra (Ankner et al., 2024) | $2.59 \pm 0.022$ | $1.98 \pm 0.006$ | $1.91 \pm 0.010$ | $\mathbf{2.16 \pm 0.011}$ | $2.56 \pm 0.031$ | $1.86 \pm 0.030$ | $2.18 \pm 0.007$ |
| Eagle (Li et al., 2024b) | $2.50 \pm 0.057$ | $1.80 \pm 0.009$ | $2.10 \pm 0.005$ | $1.93 \pm 0.028$ | $2.49 \pm 0.013$ | $1.94 \pm 0.017$ | $2.13 \pm 0.010$ |
| Recycling (Luo et al., 2025) | $2.08 \pm 0.045$ | $1.77 \pm 0.008$ | $1.93 \pm 0.004$ | $1.83 \pm 0.007$ | $2.21 \pm 0.010$ | $1.73 \pm 0.013$ | $1.93 \pm 0.007$ |
| Medusa (Cai et al., 2024) | $2.08 \pm 0.011$ | $1.69 \pm 0.006$ | $1.57 \pm 0.002$ | $1.76 \pm 0.019$ | $2.07 \pm 0.008$ | $1.55 \pm 0.015$ | $1.79 \pm 0.004$ |
| PLD (Saxena, 2023) | $1.65 \pm 0.024$ | $1.08 \pm 0.008$ | $\mathbf{2.50 \pm 0.004}$ | $1.20 \pm 0.011$ | $1.62 \pm 0.005$ | $1.75 \pm 0.006$ | $1.63 \pm 0.004$ |
| SpS (Chen et al., 2023) | $1.57 \pm 0.018$ | $1.12 \pm 0.007$ | $1.54 \pm 0.006$ | $1.39 \pm 0.008$ | $1.40 \pm 0.021$ | $1.55 \pm 0.023$ | $1.43 \pm 0.006$ |
| Lookahead (Fu et al., 2024) | $1.46 \pm 0.021$ | $1.13 \pm 0.024$ | $1.26 \pm 0.005$ | $1.24 \pm 0.010$ | $1.55 \pm 0.003$ | $1.17 \pm 0.007$ | $1.30 \pm 0.005$ |
| None | $1.00$ | $1.00$ | $1.00$ | $1.00$ | $1.00$ | $1.00$ | $1.00$ |

Table 4: **Wall-clock speedups under teacher verification** (↑). Errors indicate the standard error over three runs. Temperature is set to $0.7$. Across most tasks, O-PTP achieves the largest speed-up.

| Parallelization | MTC | TL | SUM | QA | Math | RAG | Task-Average |
|---|---|---|---|---|---|---|---|
| O-PTP, sequential (ours) | $\mathbf{4.52 \pm 0.019}$ | $\mathbf{3.73 \pm 0.041}$ | $\mathbf{4.53 \pm 0.027}$ | $3.42 \pm 0.022$ | $\mathbf{4.62 \pm 0.028}$ | $\mathbf{4.29 \pm 0.037}$ | $\mathbf{4.21 \pm 0.011}$ |
| O-PTP, parallel (ours) | $3.00 \pm 0.015$ | $2.59 \pm 0.037$ | $3.12 \pm 0.024$ | $2.23 \pm 0.017$ | $3.44 \pm 0.027$ | $2.93 \pm 0.032$ | $2.89 \pm 0.009$ |
| SAMD (Hu et al., 2025) | $\mathbf{4.52 \pm 0.023}$ | $3.16 \pm 0.032$ | $4.08 \pm 0.036$ | $\mathbf{3.44 \pm 0.028}$ | $4.37 \pm 0.024$ | $3.86 \pm 0.045$ | $3.95 \pm 0.011$ |
| Eagle-2 (Li et al., 2024a) | $4.44 \pm 0.011$ | $3.20 \pm 0.031$ | $3.79 \pm 0.017$ | $3.40 \pm 0.019$ | $4.62 \pm 0.021$ | $3.89 \pm 0.024$ | $3.93 \pm 0.008$ |
| Hydra (Ankner et al., 2024) | $3.90 \pm 0.006$ | $2.88 \pm 0.023$ | $2.95 \pm 0.010$ | $3.21 \pm 0.013$ | $3.90 \pm 0.009$ | $3.37 \pm 0.014$ | $3.40 \pm 0.005$ |
| Eagle (Li et al., 2024b) | $3.63 \pm 0.008$ | $2.72 \pm 0.024$ | $3.17 \pm 0.014$ | $2.88 \pm 0.015$ | $3.76 \pm 0.014$ | $3.14 \pm 0.018$ | $3.24 \pm 0.006$ |
| Recycling (Luo et al., 2025) | $2.74 \pm 0.007$ | $2.46 \pm 0.029$ | $2.68 \pm 0.013$ | $2.60 \pm 0.014$ | $3.06 \pm 0.014$ | $2.60 \pm 0.017$ | $2.70 \pm 0.006$ |
| Medusa (Cai et al., 2024) | $2.70 \pm 0.005$ | $2.18 \pm 0.016$ | $2.12 \pm 0.007$ | $2.24 \pm 0.009$ | $2.67 \pm 0.008$ | $2.26 \pm 0.010$ | $2.38 \pm 0.003$ |
| SpS (Chen et al., 2023) | $2.10 \pm 0.006$ | $1.40 \pm 0.009$ | $2.15 \pm 0.011$ | $1.79 \pm 0.008$ | $1.89 \pm 0.008$ | $2.17 \pm 0.014$ | $1.93 \pm 0.003$ |
| PLD (Saxena, 2023) | $1.67 \pm 0.007$ | $1.10 \pm 0.007$ | $2.73 \pm 0.024$ | $1.37 \pm 0.017$ | $1.80 \pm 0.012$ | $1.71 \pm 0.016$ | $1.73 \pm 0.005$ |
| Lookahead (Fu et al., 2024) | $1.69 \pm 0.004$ | $1.24 \pm 0.010$ | $1.54 \pm 0.006$ | $1.56 \pm 0.007$ | $1.92 \pm 0.008$ | $1.42 \pm 0.008$ | $1.57 \pm 0.003$ |
| None | $1.00$ | $1.00$ | $1.00$ | $1.00$ | $1.00$ | $1.00$ | $1.00$ |

Table 5: **Average number of accepted tokens under teacher verification** (↑). For each method and task, we list the average length of the longest prefix matching the teacher's output, which directly determines decoding speedup. Errors indicate the standard error over total number of model calls. Temperature is set to $0.7$. Across nearly all tasks, O-PTP achieves the highest number of accepted tokens. Parallel verification trades number of accepted tokens for total speedup, so we report both for our model.

## E    ADDITIONAL ABLATION RESULTS

Based on a dataset (NYC TLC, 2017) that contains latitudes and longitudes for pick-up locations for all taxi rides in 2016, we divide the city into $25$ neighborhoods via $k$-Means clustering to obtain a discrete-valued time-series that we can split into overlapping chunks of length $N$. This is a common benchmark dataset in the literature of marked temporal point processes (Xue et al., 2024), and autoregressive transformers are a common architecture (Draxler et al., 2025).

We now provide some of the specific choices we made when implementing the general framework of Parallel Token Prediction. Specifically, we discuss the empirical difference between O-PTP and C-PTP and which specific loss to choose. We will specify our model architecture and how to embed both tokens and auxiliary variables in the same embedding space, and lastly compare the proposal distributions our training sequences can be sampled from.

We test our framework by training a model that predicts pick-up locations for taxis in New York City. Based on a dataset (NYC TLC, 2017) that contains latitudes and longitudes for pick-up locations for all taxi rides in 2016, we divide the city into $25$ neighborhoods via $k$-Means clustering to obtain a discrete-valued time-series that we can split into overlapping chunks of length $N$. This is a common benchmark dataset in the literature of marked temporal point processes (Xue et al., 2024).

As a teacher model, we pretrain a 29M-parameter autoregressive causal transformer based on the architecture of GPT-2 (Radford et al., 2019), using the cross-entropy loss in Eq. (8). For our PLM

we choose the same GPT-style transformer architecture as the teacher. This allows us to use the teacher's parameters as a warm-start. We evaluate all our parallel models in terms of the average number of leading tokens predicted by our student model that are identical to the teacher. In the end, this is the quantity that limits the maximum latency reduction that can be achieved, see Section 3.

### E.1 AUXILIARY VARIABLE EMBEDDINGS

In our experiments we use transformers that embed tokens into a higher-dimensional embedding space via a learned embedding before adding a positional embedding. This doesn't work out-of-the-box for our auxiliary variables since they are one-dimensional continuous variables. Thus we learn a separate embedding. We combine two components, for each of which we test several variants: (1) A learned affine linear transform [**lin**] or a fully connected neural network [**NN**]. (2) Feed either the scalar $u$ [**fl**], an $n$-dimensional threshold-embedding $e_j = 1\{u \leq j/n\}$ [**th**], or an $n$-dimensional embedding $e_j = 1\{u2^{j-1} \bmod 1 \leq 0.5\}$ [**ar**] inspired by arithmetic coding (Witten et al., 1987).

Empirically, all methods work reasonably well, but a structured embedding leads to faster and more stable training convergence. This is similar to the transformer's positional embedding where both learned and fixed embeddings work well but the latter is preferred in practice (Vaswani et al., 2017). For further experiments we use the [**ar + lin**] embedding. Table 7 shows the detailed effect of different embedding strategies.

### E.2 DISTILLATION LOSSES AND PROPOSAL DISTRIBUTIONS

Equations (10) to (12) leave freedom as to how to choose the source of training sequences. In this section, we discuss and empirically compare several options.

If our goal is to deploy our parallelized student as a drop-in replacement of our teacher model, the lowest-variance option is to sample training sequences from the teacher. Another possibility is to directly sample training sequences from a dataset, such as the one that was used to train the teacher model in the first place. This has the advantage that we can compute the teacher predictions $Q_\varphi(t_k \mid t_{<k})$ in parallel over a full sequence instead of iteratively having to generate it. Finally, we can sample sequences directly from the student model by first sampling auxiliary variables $u_i, \ldots, u_T \sim \mathcal{U}[0,1]$ and then using our student model in its current state to sample training sequences in parallel. As the student's prediction gets closer to that of the teacher during training, this approaches the same training sequence distribution as if we had sampled the teacher directly.

Conceptually, the latter is similar to the techniques used in distilling the autoregressive WaveNet into a parallel model (Oord et al., 2018). In practice, following their approach, which equates to using a reverse KL loss as noted above, is efficient but proved to be too unstable. Parallel WaveNet stabilizes training with several auxiliary losses, like a perceptual loss, which we do not have in our setting, in general.

We now empirically compare the distillation losses (Section 2.2.1), focusing on KL and cross-entropy losses in Eqs. (10) and (12). Specifically the KL loss (**kl**), reverse KL loss (**kl-rev**), binary cross-entropy loss (**bce**), and categorical cross-entropy loss (**ce**). During training we sample training sequences from a dataset and continuations $t_{\geq i}$ either from the teacher model $Q_\varphi$, the student model $P_\theta$, or directly from a dataset. Table 6 shows the results for different losses. Empirically we note, that O-PTPs are easier to train than C-PTPs and achieve a higher number of average correct tokens. This is most likely due to the fact that O-PTPs do not have to predict the full token distribution accurately, which includes tail behavior, as long as they learn which token is the most likely given the auxiliary variable. In the following, we choose to sample training sequences from the teacher model for best results.

### E.3 SAMPLING OF AUXILIARY VARIABLES

Our framework conditions, for a prompt $t_{<i}$, not on token $t_k$ directly but on the auxiliary variable $u_k \in [F_{k,t_k-1}, F_{k,t_k}]$ that contains the same information. During inference we sample $u_k \sim \mathcal{U}[0,1]$ as to not bias our predictions. During training on the other hand, we have more flexibility and can sample the permissible interval using $u_k = F_{k,t_k-1} + \tilde{u}_k [F_{k,t_k} - F_{k,t_k-1}]$, where $\tilde{u}_k \sim \text{Beta}(b,b)$. For $b = 1$ this simplifies to a uniform distribution while $b \neq 1$ puts more or less weight on predic-

| Proposal Distribution $P(t)$ | kl | kl-rev | bce | ce | MTP |
|---|---|---|---|---|---|
| Teacher | 40 | 41 | **45** | 44 | 10.1 |
| Student | 44 | 39 | **45** | **45** | 10.1 |
| Dataset | 29 | 36 | 44 | 43 | 10.1 |

Table 6: **Our framework is compatible with several losses.** Average number of correct tokens (↑) on the taxi dataset, evaluated on 16000 samples. O-PTP are distilled with KL or reverse KL loss (**kl**, **kl-rev**), C-PTP with binary or categorical cross entropy loss (**bce**, **ce**). Independent prediction (MTP) (Gloeckle et al., 2024) achieves 10.1. Numbers rounded to reflect level of statistical certainty.

| Model | fl + NN | th + lin | th + NN | ar + lin | ar + NN | MTP |
|---|---|---|---|---|---|---|
| O-PTP | 35.9 | 40.9 | 39.1 | 45.4 | **46.1** | 10.1 |
| C-PTP | 28.8 | 36.8 | 36.6 | **40.4** | 35.7 | 10.1 |

Table 7: **Structured embeddings of auxiliary variables $u_k$ are more stable than fully-learned embeddings.** Average number of correct tokens (↑) on the taxi dataset, evaluated on 16000 samples. Trained using the KL loss (C-PTP) and binary cross-entropy loss (O-PTP), respectively. Independent prediction (MTP) (Gloeckle et al., 2024) achieves 10.1. Numbers rounded to reflect level of statistical certainty.

tions that land closer to the border of the permissible interval and thus are more difficult to predict. Training results for different values of $b$ can be found in Table 8. Empirically, we find that while the choice of $b$ does not seem to affect the final average number of correct samples, a larger $b$ might speed up the earlier stages of training while a smaller $b$ might yield slightly better sample quality during inference, as measured by model perplexity.

### E.4 Abundant Computational Resources

Another way to leverage additional compute (via additional hardware or longer predictions) in a pure inference setting is to improve the expected number of correct tokens directly. Specifically, for a fixed context we can compute $M$ independent predictions using independently drawn auxiliary variables $u_{i,m}, \ldots, u_{N,m}$ for $m = 1, \ldots, M$. By choosing the best prediction, i.e. the one that gives us the best chance of a higher number of correct tokens, we can improve latency further.

Crucially, we have to choose the best prediction in a way that doesn't bias the marginal distribution over future tokens. If we, for example, naively choose the sequence that is correct for the most amount of tokens, we will bias our prediction towards sequences that are easier to predict. One way to achieve bias-free improvements is to pick the set of auxiliary variables that lands, on average, closest to the center of a token's valid interval $I_k(t_k) = [F_{k,t_k}, F_{k,t_k-1})$ where $F_{k,t_k}$ is the cumulative probability under $Q_\varphi$ to choose $t_k$ when predicting that token. Specifically, choose

$$\text{argmax}_m \sum_{k=i}^{N} \left| \frac{u_{k,m} - F_{k,t_{k,m}}}{F_{k,t_{k,m}-1} - F_{k,t_{k,m}}} - \frac{1}{2} \right|. \tag{22}$$

This does not bias the marginal distribution but does bias the distribution of the selected $u_k$ to be closer to the center of its interval $I_k(t_k)$, making the prediction less prone to small differences in

| $b$ | 2 | 1 | 0.5 | MTP |
|---|---|---|---|---|
| O-PTP | 12.9 | **13.9** | 13.8 | 8.4 |
| C-PTP | 13.6 | **13.8** | 13.5 | 8.4 |

Table 8: **Different sampling strategies for $u_k$ are available.** Average number of correct tokens (↑) for $\tilde{u}_k \sim \text{Beta}(b, b)$ on the taxi dataset, evaluated on 16000 samples, with $N = 16$. Trained using the KL loss (C-PTP) and binary cross-entropy loss (O-PTP), respectively. Independent prediction (MTP) (Gloeckle et al., 2024) achieves 8.4. Numbers rounded to reflect level of statistical certainty.

the teacher's and student's logits. In the limit $M \to \infty$ we always select the middle point of $I_k(t_k)$, yielding an upper bound to the possible improvement. Table 9 shows the performance gains on the taxi dataset.

| M | 1 | 4 | 16 | 64 | 256 | 1024 | $10^6$ | $\infty$ |
|---|---|---|----|----|-----|------|--------|----------|
| Avg. correct tokens | 45.36 | 49.67 | 54.76 | 55.74 | 56.91 | 59.79 | 65.42 | 90.17 |

Table 9: **Additional compute increases correctness**. Average number of correct tokens ($\uparrow$) for $M$ predicted sequences, computed in parallel on the taxi dataset, with sequence length $N = 100$.

### E.5 RESTRICTED COMPUTATIONAL RESOURCES

Limiting the number $N$ of tokens our PTP predicts at once to a smaller number will reduce the total number of floating point operations, increasing energy efficiency. This, of course, negatively affects the possible latency gains, especially since $N$ is an upper bound on the average number of correct tokens. Table 10 shows the result for different values of $N$ on the taxi dataset.

| N | 1 | 2 | 4 | 8 | 16 | 64 | 100 | $\infty$ |
|---|---|---|---|---|----|----|-----|----------|
| Avg. correct tokens | 1.00 | 1.99 | 3.93 | 7.59 | 13.90 | 36.60 | 45.36 | 48.76 |
| MTP | 1.00 | 1.91 | 3.53 | 5.86 | 8.40 | 10.08 | 10.07 | 10.20 |

Table 10: **Less compute decreases correctness**. Average number of correct tokens ($\uparrow$) for limited number of predicted tokens $N$ per O-PTP call on the taxi dataset, $M = 1$.

## F EFFICIENT ERROR CORRECTION

Algorithm 1 shows how to correct errors with sequential calls to a proposal model and a base model.

Algorithm 2 shows how to correct errors by running a proposal model and a base model in parallel. This can also be achieved in a single model call when leveraging Gated LoRA (Samragh et al., 2025): Algorithm 2 can be executed in a single forward pass by structuring the attention mask as shown in Fig. 10: the verification rows attend only to the prefix (base model weights active), while each proposal row attends to its hypothetical prefix plus all preceding proposal tokens (student weights active), enabling the base and student computations to run in parallel within one model call.

---

**Algorithm 1** Sampling with sequential error correction

---

**Require:** Prompt $t_{<i}$, one-hot or categorical PTP $P_\theta$, verification model $Q_\varphi$, target length $T$.

   Sample $u_k \sim \mathcal{U}[0, 1]$ for all $k \geq i$.
   **while** $i < T$ **do**
      **if** $P_\theta$ is one-hot PTP **then**
         $P_k \leftarrow P_\theta(t_k \mid t_{<i}, u_i, \ldots, u_k)$, jointly for all $k \geq i$     ▷ Student one-hot distributions
         $t_k \leftarrow \mathrm{argmax}_l P_{kl}$ for all $k \geq i$     ▷ Eq. (5)
      **else**
         $P_k \leftarrow P_\theta(t_k \mid t_{<i}, u_i, \ldots, u_{k-1})$, jointly for all $k \geq i$   ▷ Student categorical distributions
         $t_k = \mathrm{Pick}(u_k, P_k)$ for all $k \geq i$     ▷ Eq. (7)
      **end if**
      $Q_k \leftarrow Q_\varphi(t_k \mid t_{<k})$, jointly for all $k \geq i$     ▷ Teacher categorical distributions
      $\tilde{t}_k \leftarrow \mathrm{Pick}(u_k, Q_k)$ for all $k \geq i$     ▷ Teacher tokens
      $i \leftarrow \min_{k \geq i}\{k : t_k \neq \tilde{t}_k\}$     ▷ cf. Eq. (20)
      $t_i \leftarrow \tilde{t}_i$
      $i \leftarrow i + 1$     ▷ #accepted = #correct + 1
   **end while**

---

---

**Algorithm 2** Sampling with Partial Quadratic Decoding

---

**Require:** Prompt $t_{<i}$, one-hot PTP $P_\theta$, verification model $Q_\varphi$, token-budget $C$, utility function $H$,
    target length $T$.
    Sample $u_k \sim \mathcal{U}[0,1]$ for all $k \geq i$.
    $t_k \leftarrow t_{\text{pad}}$ for all $k \geq i$
    $c_k \leftarrow 0$ for all $k \geq i$                 $\triangleright$ Only allocate to $m = 0$ in first iteration
    **while** $i < T$ **do**
        Compute optimal allocation $B$ of $C$ tokens from $c_k$ under $H$.        $\triangleright$ Eq. (15)
        $P_{mk} \leftarrow P_\theta(t_{k+m} \mid t_{<i+m}, u_{i+m}, \ldots, u_{k+m})$, jointly for all $i + B_m > k \geq i, m \geq 0$
        $t_{mk} \leftarrow \arg\max_l P_{mkl}$                       $\triangleright$ Eq. (5)
        $c_{mk} \leftarrow \max_l P_{mkl}$                     $\triangleright$ Proposal Confidence
        $Q_k \leftarrow Q_\varphi(t_k \mid t_{<k})$, jointly for all $k \geq i$       $\triangleright$ $Q_\varphi$ and $P_\theta$ can be run in parallel
        $\tilde{t}_k \leftarrow \text{Pick}(u_k, Q_k)$ for all $k \geq i$
        $m \leftarrow \min_{k \geq i}\{k : t_k \neq \tilde{t}_k\} - i$               $\triangleright$ cf. Eq. (20)
        $t_{k+m} \leftarrow t_{mk}$ for $i + B_m > k \geq i$
        $c_{k+m} \leftarrow c_{mk}$ for $i + B_m > k \geq i$
        $i \leftarrow i + m$
        $t_i \leftarrow \tilde{t}_i$
        $i \leftarrow i + 1$                        $\triangleright$ #accepted $=$ #correct
    **end while**

---

| AR Step 1 | AR Step 2 | AR Step 3 | AR Step 4 | AR Step 5 |
|---|---|---|---|---|
| def | def factor | def factorial | def factorial( | def factorial(num |

| PTP Step 1 | PTP Step 2 | PTP Step 3 | PTP Step 4 | PTP Step 5 |
|---|---|---|---|---|
| def factorial(n): \n if n == 0: \n return : \n | def factorial(num): \n if n == 0: \n return 1 \n else: \n | def factorial(num): \n if num <= 1: \n return 1 \n else: \n return factor \n \n | def factorial(num): \n if num <= 1: \n return 1 \n else: \n return num * factorial(num-1 1 | def factorial(num): \n if num <= 1: \n return 1 \n else: \n return num * factorial(num-1) |

Figure 8: Detailed generation trajectory for Fig. 1. Green is accepted, red is rejected by the base model call in the next step.

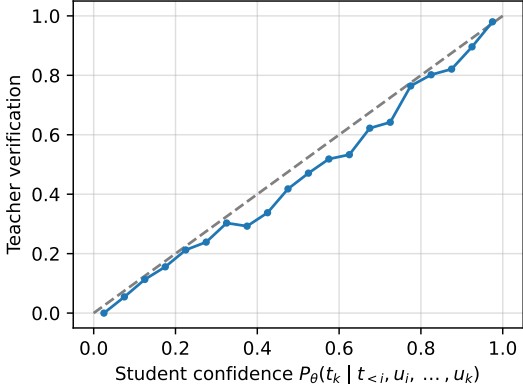

Figure 9: The learned probabilities $P_\theta(t_k \mid t_{<i}, u_i, \ldots, u_k)$ of One-Hot Parallel Token Prediction (O-PTP) are reliable estimators as to whether a token will be accepted upon teacher verification.

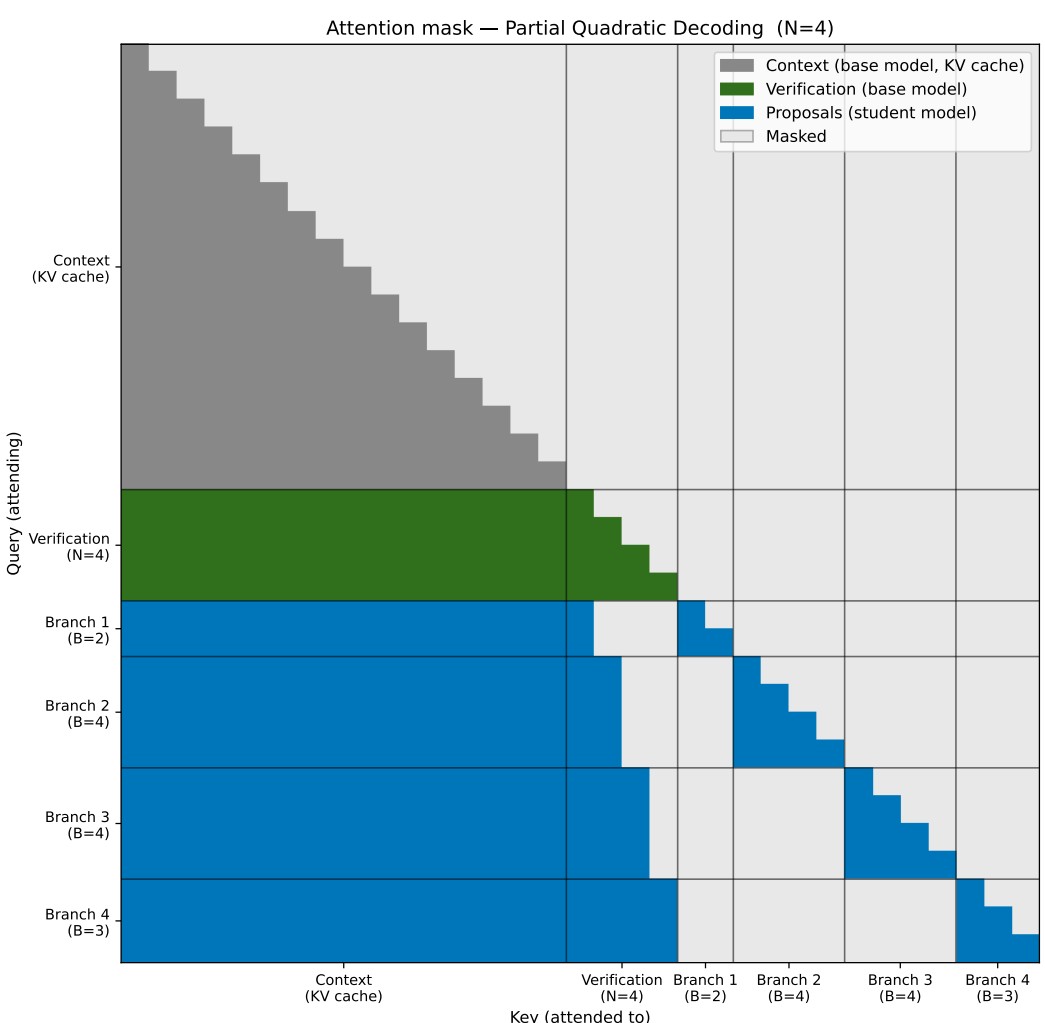

Figure 10: Attention mask used for efficient parallel decoding, when the PTP model is adapted from its base model via gated LoRA (Samragh et al., 2025). The attention mask corresponds to Fig. 5. The color in each row indicates what model weights are active: not active because contained in KV cache, base model for verification, and student model for proposals.

# G  Experimental Details

## G.1  Training Algorithms

Algorithm 3 shows how to distill a PTP from a teacher, Algorithm 4 shows how to train directly from data.

---

**Algorithm 3** Training PTP (distillation)

---

**Require:** Sequence proposal distribution $P(t)$ (teacher, student, dataset, or combination), cutoff distribution $P(i \mid t)$, teacher model $Q_\varphi(t)$, one-hot or categorical PTP $P_\theta$.
   **while** not converged **do**
      Sample $t \sim P(t)$
      Sample $i \sim P(i \mid t)$.
      Choose $N$ such that $i < N \le T$.
      $P_k = Q_\varphi(t_k \mid t_{<k})$ in single model call for all $k = i, \ldots, N$.
      Sample $u_k \in [F_{k,t_k-1}, F_{k,t_k})$ for all $k = i, \ldots, N$.
      Compute $\nabla_\theta \mathcal{L}(\theta, t)$ using one of Eqs. (10) to (12).
      Gradient step.
   **end while**

---

**Algorithm 4** Training PTP (inverse autoregressive)

---

**Require:** Dataset $P(t)$, cutoff distribution $P(i \mid t)$, categorical PTP $P_\theta$.
   **while** not converged **do**
      Sample training sequence $t_1, \ldots, t_T$.
      Sample split position $i \in \{1, \ldots, T\}$.
      Choose $N$ such that $i < N \le T$.
      **for** $k = i, \ldots, N$ **do**
         $P_k = P_\theta(t_k \mid t_{<i}, u_i, \ldots, u_{k-1})$, with auxiliary available from previous iterations.
         Sample $u_k \in [F_{k,t_k-1}, F_{k,t_k})$.
      **end for**
      Compute $\nabla_\theta \mathcal{L}(\theta, t)$ using Eq. (13).
      Gradient step.
   **end while**

---

## G.2  Training Details

The model used in Section 4.1 and Appendix E is a GPT-2–style transformer language model with 4 transformer layers, a hidden size of 1536, and approximately 29 million trainable parameters. Each layer follows the standard GPT-2 architecture, consisting of multi-head self-attention and position-wise feedforward sublayers, combined with residual connections and layer normalization. The vocabulary size is set to 25. Unless otherwise noted, all other hyperparameters and initialization schemes follow the original GPT-2 specification (Radford et al., 2019). During training and inference of our student model we don't provide any context and evaluate the correctness of the next $N = 100$ tokens, by comparing $Q_\varphi(t_k \mid t_{<k})$ and $P_\theta(t_k)$. For results on a smaller $N = 16$, see Appendix E.5. We train every model for 150k steps with a batch size of 32 with the Adam optimizer (Kingma & Ba, 2015) and learning rate 0.0001.

The teacher model used in Section 4.2 is a dialogue-tuned variant of the TinyLlama (Zhang et al., 2024) 1.1 billion parameter model, adopting the same architecture and tokenizer as LLaMA 2 (Touvron et al., 2023): `TinyLlama-1.1B-Chat-v1.0`. The model uses a transformer architecture comprising 22 transformer layers, each with standard multi-head self-attention, SwiGLU feedforward blocks, residual connections, and layer normalization. The embedding and hidden dimension is 2048, and the intermediate (feedforward) dimension is 5632, consistent with a LLaMA-style scaling. The vocabulary size is $32,000$. The parameters are available via `https://huggingface.co/TinyLlama/TinyLlama-1.1B-Chat-v1.0`.

We train an O-PTP on predicting 16 additional tokens. We adopt the strategy of finetuning using gated low-rank adaptation, where changed parameters are only applied for completion to-

kens (Samragh et al., 2025). We choose a LoRA (Hu et al., 2022) rank of $r = 256$, although comparable speedups can be achieved already with $r = 64$. We train every model for 1.3M steps with a batch size of 56 with the AdamW optimizer (Loshchilov & Hutter, 2019) on Eq. (10) and learning rate 0.0001. We generate training and validation data by generating Python code completions on CodeContests (Li et al., 2022) from the teacher, splitting generated sequences randomly into input and completion. We use a teacher sampling temperature of 0.7, top-$k = 50$ and top-$p = 0.9$, as is recommended for this model.

For the MTP baseline, we use Eq. (11) with uninformative $u$s in otherwise identical code for a fair comparison.

For the scaling experiments in Section 4.2.1, we parameterize models of different scales as in Table 11. We adopt the same architecture choices as TinyLlama.

For the natural language tasks on SpecBench (Xia et al., 2024), we finetune Vicuna-7B-v1.5 (Chiang et al., 2023) on ShareGPT (Chen et al., 2024). We train it with the same gated $r = 128$ LoRA setup as for TinyLlama, but with batch size 64 and for 1M steps.

| Model Size | Hidden Size | Hidden Layers | Attention Heads |
|---|---|---|---|
| 4k | 16 | 1 | 1 |
| 66k | 64 | 1 | 1 |
| 525k | 128 | 2 | 2 |
| 4.2M | 256 | 4 | 4 |
| 34M | 512 | 8 | 8 |
| 268M | 1024 | 16 | 16 |
| 1.1B (teacher) | 2048 | 22 | 32 |

Table 11: Model configuration per parameter count for scaling experiment

We base our code on `transformers` (Wolf et al., 2019), `PyTorch` (Paszke et al., 2019), PyTorch Lightning (Falcon & The PyTorch Lightning team, 2019), Numpy (Harris et al., 2020), Matplotlib (Hunter, 2007) for plotting and Pandas (McKinney, 2010) for data evaluation.

## H  SELF-VERIFICATION

The experiments in Section 4.3 use a separate teacher model for verification, guaranteeing output identical in distribution to the teacher. Here we explore a faster alternative that requires no teacher at all: the PTP student model verifies its own proposals.

After drafting tokens with auxiliary variables $u_k$, the student evaluates its marginal distribution $P_\theta(\cdot \mid t_{<k})$ with the gated adapter weights active but *without* any auxiliary variable input. Per Theorem 2, this recovers the standard autoregressive distribution of the student. A proposed token $t_k$ is accepted if it matches $\text{Pick}(u_k, P_\theta(t_k \mid t_{<k}))$. Since both proposal and verification use the same model in a single forward pass, no separate teacher call is required.

Under self-verification on SpecBench, the average number of accepted tokens is #accepted $= 6.1$, compared to #accepted $= 4.2$ under teacher verification (Table 3). The higher acceptance rate arises because the student's drafts are already well-aligned with its own marginal distribution. This translates to a significantly higher potential speedup than what is reported in the main text, at the cost of no longer guaranteeing identity with the teacher's output distribution.

Below, we show uncurated outputs from SpecBench prompts (Xia et al., 2024). Despite the absence of teacher verification, the student produces coherent, instruction-following responses.

### H.1  CODING

**Prompt:** Write a C++ program to find the nth Fibonacci number using recursion.

**Response:**

```
1  Here is a simple program that finds the nth Fibonacci number using
       recursion in C++:
2  ```
3  #include <iostream>
4
5  using namespace std;
6
7  int fibonacci(int n) {
8      if (n <= 0) {
9          return 0;
10     } else if (n == 1) {
11         return 1;
12     } else if (n == 2) {
13         return 2;
14     } else {
15         return fibonacci(n-1) + fibonacci(n-2);
16     }
17 }
18
19 int main() {
20     int n;
21     cout << "Enter the number of Fibonacci numbers to find: ";
22     cin >> n;
23     cout << "The first " << n << " Fibonacci numbers are:\n";
24     for (int i = 0; i < n; i++) {
25         cout << "Fibonacci(" << i << ") = " << fibonacci(i) << endl;
26     }
27     return 0;
28 }
29 ```
30 This program asks the user for the number of Fibonacci numbers they want
       to find, and then prints out the first n Fibonacci numbers. The
       function `fibonacci` is called recursively to compute the nth
       Fibonacci number, with the base cases being 0, 1, and 2. The `main`
       function is used to handle the user input and print out the results.
31
32 Note: This is a very simple implementation of the Fibonacci sequence, and
        it may not be suitable for larger values of n. For larger values of
       n, you may need to use a different algorithm, such as the Golden
       Ratio algorithm or the Matrix Exponential algorithm. Additionally,
       this implementation assumes that the input value is positive, so you
       may need to handle negative input values appropriately.
```

## H.2 EXTRACTION

**Prompt:** Evaluate the following movie reviews on a scale of 1 to 5, with 1 being very negative, 3 being neutral, and 5 being very positive: 1. This movie released on Nov. 18, 2019, was phenomenal. The cinematography, the acting, the plot - everything was top-notch. 2. Never before have I been so disappointed with a movie. The plot was predictable and the characters were one-dimensional. In my opinion, this movie is the worst one to have been released in 2022. 3. The movie was okay. There were some parts I enjoyed, but there were also parts that felt lackluster. This is a movie that was released in Feb 2018 and seems to be quite ordinary. Return the answer as a JSON array of integers.

**Response:**

```
1  [1, 4, 3]
```

## H.3 MATH

**Prompt:** When rolling two dice, what is the probability that you roll a total number that is at least 3?

**Response:**

```
1  There are two ways to roll a total number that is at least 3, 3 and 3, or
       4 and 2. So the probability is 1/36.
2
3  To calculate this, you can use the formula for combinations, which is:
4
5  n(n+1)/2 = P(A)
6
7  where n is the number of choices, and P(A) is the probability of A
       occurring. In this case, n = 2, so the probability of rolling a total
       number that is at least 3 is 1/2\*1/2 = 1/36.
8
9  Alternatively, you can use the formula for the binomial distribution,
       which is:
10
11 P(A) = (n! / (n-k)!) \* (1-p)^k \* p^n
12
13 where n is the number of trials, k is the number of successes, and p is
       the probability of success. In this case, n = 2, k = 1, and p = 1/36.
        Plugging these values into the formula, we get P(A) = (2! / (2-1)!)
       \* (1-1/36)^1 \* 36/36 = 1/36.
14
15 I hope this helps! Let me know if you have any more questions.
```

## H.4 MATH REASONING

**Prompt:** A company wanted to buy 500 computers and had $700 for each computer. However, when they went to order, they realized the price of each computer was 10% higher than they had planned for. Calculate the total amount of money they paid to buy the computers at the new prices.

**Response:**

```
1  The price of each computer was 10% higher than planned, so the new price
       of each computer was 700 \* 1.10 = $770.
2  The company wanted to buy 500 computers, so they paid $770 \* 500 =
       $38500 for the computers.
3  Therefore, the company paid $38,500 to buy the computers at the new
       prices.
4  It's important to note that this is just an example, and the actual
       amount of money paid by the company would depend on the specifics of
       the purchase, such as any discounts or taxes that may have been
       applied. The calculation provided here is just a general example of
       how to calculate the total amount of money paid for the computers at
       the new prices.
```

## H.5 QA

**Prompt:** Name of the last part of harry potter?

**Response:**

```
1  The last part of Harry Potter is "Harry Potter and the Deathly Hallows,
       Part 2." It was released in 2007 and is the final installment in the
       Harry Potter series by J.K. Rowling. The film adaptation of the book
       was directed by David Yates and starred Daniel Radcliffe, Emma Watson
       , and Rupert Grint. The Deathly Hallows Part 2 follows the story of
       Harry Potter and his friends as they confront the dark lord Voldemort
        and the Deathly Hallows, a powerful and dangerous object that grants
        its possessor immense power. The film was a critical and commercial
       success, and it brought an end to the Harry Potter series, which had
       captivated audiences worldwide for over a decade. The series as a
       whole is considered one of the most successful and influential book
```

```
      and film series in recent history, and its legacy continues to be
      felt in popular culture.
```

## H.6 REASONING

**Prompt:** One morning after sunrise, Suresh was standing facing a pole. The shadow of the pole fell exactly to his right. Can you tell me the direction towards which the shadow was pointing - east, south, west, or north? Explain your reasoning steps.

**Response:**

```
1  To determine the direction towards which the shadow was pointing, we need
       to use the relationship between the angle of the shadow and the
       angle of the sunrise.
2
3  Firstly, we know that the sun rises in the east and sets in the west. So,
       if the shadow is pointing towards the east, then the pole is
       pointing towards the west.
4
5  Now, we need to determine the angle between the shadow and the pole.
       Since the shadow is pointing towards the east, we can draw an arrow
       from the shadow to the east and another arrow from the pole to the
       west.
6
7  The angle between these two arrows will give us the angle of the shadow.
       Since the shadow is pointing towards the east, the angle between the
       arrow and the pole will be a right angle, or 90 degrees.
8
9  Therefore, the direction towards which the shadow was pointing was east.
10
11 In summary, the pole was pointing towards the east, and the shadow was
       pointing towards the east as well. The angle of the shadow was 90
       degrees, which means the pole and the shadow were in line with each
       other. So, the direction towards which the shadow was pointing was
       east.
```

## H.7 ROLEPLAY

**Prompt:** Embrace the role of Sheldon from "The Big Bang Theory" as we delve into our conversation. Don't start with phrases like "As Sheldon". Let's kick things off with the following question: "What is your opinion on hand dryers?"

**Response:**

```
1  As Sheldon, I have a bit of a love-hate relationship with hand dryers. On
       one hand, I find them to be quite efficient and convenient,
       especially when compared to paper towels or cloth towels. However, on
        the other hand, I feel that they can be quite loud and disruptive,
       especially in public restrooms. Overall, I think they are a decent
       choice for drying your hands, but it's important to be mindful of
       others around you and try not to use them in situations where they
       may be a nuisance.
```

## H.8 STEM

**Prompt:** How have the Alps and Rhine River influenced settlement and agriculture in Western Europe? List three impacts.

**Response:**

```
1  The Alps and Rhine River have had a significant impact on settlement and
      agriculture in Western Europe. Here are three ways in which this has
      occurred:
2
3  1. Climate: The Alps act as a natural barrier, blocking the flow of warm
      air and creating a cooler climate in the region. This has made it
      difficult to grow certain crops, such as wheat, which requires a
      warmer climate. However, the cooler climate has also made it easier
      to grow other crops, such as fruit and vegetables, which thrive in
      cooler temperatures.
4  2. Soil: The Rhine River has created a fertile soil deposit along its
      banks, making it an ideal location for agriculture. The river also
      provides a source of water for irrigation, which helps to support the
       growth of crops.
5  3. Trade: The Rhine River has been an important trade route for centuries
      , connecting Western Europe with other parts of the continent. This
      has made it easier for farmers in the region to sell their crops and
      has helped to support the growth of settlements along the river. The
      Alps, on the other hand, have created natural barriers that have made
       it more difficult to travel and trade goods. However, the presence
      of the mountains has also made it easier for farmers to protect their
       crops from invasions and other threats.
6
7  Overall, the Alps and Rhine River have had a significant impact on
      settlement and agriculture in Western Europe, shaping the region's
      climate, soil, and trade routes, and influencing the types of crops
      that can be grown and the settlements that can be built. These
      factors have contributed to the development of distinctive
      agricultural and settlement patterns in the region.
```

## H.9 TRANSLATION

**Prompt:** Translate German to English: Nicht zu vergessen die richtige Kosmetik und Nagelpflege .

**Response:**

```
1  Don't forget about the right cosmetics and nail care.
```

## H.10 WRITING

**Prompt:** Compose an engaging travel blog post about a recent trip to Hawaii, highlighting cultural experiences and must-see attractions.

**Response:**

```
1  Title: A Hawaii Adventure: Exploring the Islands' Rich Culture and
      Natural Beauty
2
3  Introduction:
4  Hawaii is a tropical paradise that has captivated the hearts of many
      travelers with its stunning natural wonders, crystal-clear waters,
      and lush landscapes. But what sets Hawaii apart from other vacation
      spots is its rich culture and history. In this blog post, we'll take
      a journey through the islands' cultural experiences and must-see
      attractions, leaving you with a deeper appreciation for the land of
      the setting sun.
5
6  Day 1: Oahu
7
8  * Visited the USS Arizona Memorial at Pearl Harbor, a solemn reminder of
      the tragic event that led to World War II.
9  * Explored the Nuuanu Pali Trail, offering breathtaking views of the
      island and the surrounding ocean.
```

```
10  * Enjoyed a traditional Hawaiian luau dinner, where we learned about
        Hawaiian culture, history, and traditions.
11  Day 2: Maui
12
13  * Visited the Hanauma Bay Nature Preserve, a stunning marine sanctuary
        where we could swim and snorkel among colorful fish and vibrant coral
        reefs.
14  * Explored the iconic Road to Hana, stopping at picturesque viewpoints
        and waterfalls along the way.
15  * Went whale watching and witnessed the majestic creatures in their
        natural habitat.
16
17  Day 3: Kauai
18
19  * Explored the Napali Coastline, a stunning series of cliffs, waterfalls,
        and beaches that offer breathtaking views of the ocean.
20  * Visited the Hanalei Bay, where we could swim, sunbathe on the beach,
        and watch the sunset.
21  * Enjoyed a sunset sail, where we could watch the sunset while enjoying
        the cool ocean breeze.
22  Day 4: Big Island
23
24  * Visited the Hawaii Volcanoes National Park, where we could see the
        Kilauea Volcano and explore the park's lush rainforests and
        waterfalls.
25  * Visited the Akaka Falls State Park, where we could see the stunning
        waterfall and its surrounding natural beauty.
26  * Enjoyed a farm tour and tasting at a local organic farm, where we could
         learn about sustainable farming practices and enjoy delicious fresh
        produce.
27
28  Conclusion:
29  Hawaii is more than just a vacation spot, it's a cultural and historical
        adventure. From the USS Arizona Memorial to the Akaka Falls State
        Park, there is something for everyone in Hawaii. Whether you're a
        nature lover or a cultural enthusiast, Hawaii has something special
        to offer. So pack your bags, grab your camera, and get ready for an
        adventure that you'll never forget.
```

