# OpenReview forum: "Parallel Token Prediction for  Language Models"
_ICLR.cc/2026/Conference — ICLR 2026 Poster_

### Official Review · Reviewer_5jCL · 2025-10-30

**Soundness:** 4
**Presentation:** 3
**Contribution:** 3
**Rating:** 6
**Confidence:** 3

**Summary:**

This paper introduces a novel architecture for parallel token generation, says $K$ next token simultaneously. Conventional large language models (LLMs) generate text in an autoregressive manner—given $T$ input tokens, the model predicts the probability distribution of the next token, then repeats this process sequentially. In contrast, the authors observe that the sampling step can be reformulated: rather than drawing directly from a categorical distribution, one can apply the inverse cumulative distribution function (CDF) to a uniform random variable, enabling a more parallelizable generation procedure. So, given a $K$ uniform random variable as input to the model (hypothetically), we can fully determine the next $K$ tokens without the need to perform autoregressive sampling. This paper then presents a training paradigm to train such architecture: input $T$ tokens and $K$ uniform random variables, outputs the next $K$ tokens.

**Strengths:**

Up to my knowledge, the idea presented in this work is original and interesting, drawing inspiration from the probability/sampling theory.

The paper distinguished itself clearly from other multi-token generation frameworks, including speculative decoding and discrete diffusion model. Though I'm not an expert in discrete diffusion, I think the independent generation issues of prior approach detailed in section 2.4 is an important contribution.

**Weaknesses:**

No wall-clock time experiment/ speed comparison reported.

The discussion on model architecture should be emphasized more.

It is a bit unclear what is the tradeoff between the number parallel token to be generated affects the performance. A small graph showing this would strengthen the paper.

The error correction part (section 2.3) seems to say to the use of this framework is to serve as a draft model in speculative decoding (correct me if I'm wrong). If this is the case, there should be an ablation on this.

Overall, I believe this is a neat idea. There might be some challenges training such model so I suggest the experiments to be more focused instead of spreading over multiple separate issues.

**Questions:**

In section 3.1, why are we using the taxi dataset instead of other standard dataset for finetuning LLM such as GSM8K for ablation?

Is there any tradeoff between the number parallel token to be generated and the downstream performance?

Suppose I would like to do greedy sampling or changing the sampling temperature, is there any way to do this with your framework? Is there anyway to compute the uncertainty in this framework?

Is it correct to say that the C-PTP framework can let us recover this information?


Section 2.3 seems to suggest that the model can produces mismatch between the generated tokens and the actual probability. The error correction part looks very similar to the speculative decoding process, is there anyway to get rid of this?

---

> ### Author Response · Authors · 2025-11-22
>
> We thank the reviewer for the detailed and constructive feedback and helpful suggestions.
>
> ***
>
> > No wall-clock time experiment/speed comparison
> > The error correction part (section 2.3) seems to say [PTP] serves as a draft model in speculative decoding. [...] there should be an ablation on this.
>
> Thanks for the suggestions! We add a scaling experiment that compares our PTP framework with speculative decoding on a variety of model sizes (66k to 1.1B), focusing on wall clock speedups.
>
> This figure at https://imgur.com/a/zlxbfxG shows how our PTP draft models achieve significantly higher speedups compared to same-size autoregressive draft models. This is because our models are able to predict more than one correct token in each forward pass. Note that the bigger models (268M and greater) could still improve if trained for longer.
>
> > Why are we using the taxi dataset [...] for ablation?
>
>
> We chose this dataset because it trains quickly and provides reliable hyperparameter signals. If desired, we can run ablations on the coding dataset.
>
> > [how] to do greedy sampling or changing the sampling temperature, [...] with your framework?
>
>
> Yes, this is fully supported: the error-correction step guarantees agreement with the teacher regardless of temperature. Experimentally, we find that performance decreases because there is a mismatch between student and teacher:
>
> https://imgur.com/a/JKcjnk0
>
> This can be fixed by training a student for the desired target temperature, or amortizing over temperatures. We leave such experiments open to future work.
>
> > Is there any way to compute the uncertainty in this framework?
>
> Great suggestion! For O-PTP, we predict a token probability for each token $t_i$ as a function of its $u$. Not only do we get a signal of which token $t_i$ is picked (the one with the highest probability), but we can also make direct statements about its uncertainty. For example, you can see in this plot how the model is most certain away from the edges of its bin $[F_{i,t_i}, F_{i,t_{i+1}})$:
>
> https://imgur.com/a/uV8X93L
>
> > Is it correct to say that the C-PTP framework can let us recover this information?
>
>
> Yes. C-PTP learns to approximate the teacher’s token distribution for each future token.
>
> > The error correction part [Section 2.3] looks very similar to the speculative decoding process, is there any way to get rid of this?
>
> Theoretically yes, since our framework is universal in the sense that all information needed to uniquely identify the correct sequence is available (Theorems 1 and 2). Compare this to alternative speedup methods, that either reduce transformer capacity or assume token independence. We don’t need to reduce model size to get a speedup. For example, even our 1.1B parameter PTP model achieves a wallclock speedup of 2.5x over calling the equally sized teacher model.
> Practically speaking, we get faster generation if we also do error correction. Accepting a fixed number $M$ of tokens, without verifying them, leads to inconsistent generations for larger $M$ (e.g., `def numpy`). Using error correction allows us to accept more tokens per student call (we observe 5 correct tokens on average), while retaining high quality samples.
> There are some architectural tricks such as gated LoRA [2] that could allow us to achieve a better speedup by merging the verification and prediction calls. Since our core contribution lies in the replacement of autoregressive sampling, we leave such architectural tricks to future work.
>
> [2]  Samragh et al. “Your LLM Knows the Future: [...]” arXiv:2507.11851
>
> >  It is a bit unclear what is the tradeoff between the number parallel token to be generated affects the performance. A small graph showing this would strengthen the paper.
>
>
> Great suggestion! In terms of wall-clock # tokens / second, we find that it is helpful to increase the number of tokens to be generated, but gives diminishing returns. On the coding task:
>
>
> |Requested tokens|1|2|4|8|12|16|
> |-|-|-|-|-|-|-|
> |Correct tokens|0.9|1.6|2.6|4.3|4.9|5.1|
>
> > There might be some challenges training such model
>
> Training is stable and yields strong speed-ups. Our framework is as easy to integrate as other multi-token prediction models and complements existing LLM acceleration techniques. For example, auxiliary variables can replace fixed [MASK] tokens in any other framework, providing more information to the transformer and increasing speed-ups.
>
> > The discussion on model architecture should be emphasized more.
>
> Yes, we agree. Appendix section F.2 contains a detailed discussion of the model architecture and training hyperparameters. We will further improve our paper by highlighting key points in the main text and extending the discussion in the appendix, by further details, for example including the additional experiments we have conducted based on your advice.
>
> ***
>
> We again thank the reviewer for the feedback and look forward to the subsequent discussion.

---

> > ### Comment · Reviewer_5jCL · 2025-11-22
> >
> > Thanks for the reply. I wasn’t aware of the prior works dispute raised in another review but I updated my score to 8. Good luck.

---

### Official Review · Reviewer_Efeg · 2025-11-01

**Soundness:** 2
**Presentation:** 2
**Contribution:** 1
**Rating:** 2
**Confidence:** 5

**Summary:**

The paper introduces a framework called Parallel Token Prediction (PTP) to accelerate the inference speed of autoregressive language models like transformers. The authors' key insight is to reframe the generation process: instead of the model predicting a probability distribution from which a token is then sampled, they make the sampling process itself part of the model.

By feeding a sequence of random numbers (called auxiliary variables) as input along with the text prompt, the model learns to deterministically convert these random numbers into a sequence of multiple output tokens in a single forward pass. This breaks the strict one-by-one generation bottleneck and allows for significant parallelization without making the limiting independence assumptions that plagued previous parallel decoding methods.

**Strengths:**

On a coding dataset, they distill a TinyLlama-1.1B model. Their resulting PTP model can generate an average of 5.1 correct tokens that are identical to the teacher's output, all within a single forward pass. This represents a 5x speedup over the standard autoregressive baseline.
They show that their method significantly outperforms naive parallel methods that assume token independence, which produce invalid token pairs over 60% of the time in their example.

**Weaknesses:**

Novelty: This approach has been adopted by a few other works like:
1. BiTA: Bi-Directional Tuning for Lossless Acceleration in Large Language Models
2. Your LLM Knows the Future: Uncovering Its Multi-Token Prediction Potential
I believe the authors are doing the same thing here.

Typically such multi-token prediction tasks are evaluated on Spec-Bench. These results are necessary to justify real gains at inference time.

**Questions:**

1. How do you guarantee accuracy of the teacher model during PTP?

---

> ### Author Response · Authors · 2025-11-22
>
> We thank the reviewer for the constructive feedback and helpful suggestions.
>
> ***
>
> > This approach has been adopted by a few other works like: BiTA [and] Your LLM Knows the Future.
>
> We respectfully disagree. They are important contributions, but they differ from PTP in key ways, detailed below. We will add them as related work.
>
> In detail, these methods do not incorporate auxiliary variables that uniquely determine future tokens and thus do not take full advantage of token dependencies. This also means that many of their contributions are complementary to ours - and combining both techniques has the potential to further improve the state-of-the-art. Specifically,
>
> **BiTA** proposes to learn multiple independent tokens at once using fixed [MASK] tokens. The independence assumption is a limitation we discuss in Section 2.4 and overcome with our method: In our experiments on coding tasks our model doubles (ours: 5.1 average correct tokens, independent: 2.5) the potential speed-up obtained from just independent prediction alone. BiTA mainly considers greedy prediction tasks, in which future tokens are selected deterministically and our auxiliary variables are not needed. We will clarify this fact in Theorem 1 and refer to the paper put forward by the reviewer. Fully complementary and compatible contributions of BiTA include efficient finetuning using [PROMPT] tokens, and efficient tree-based decoding of future tokens.
>
> **Your LLM Knows the Future** also proposes to learn multiple independent tokens at once using fixed [MASK] tokens. They correctly identify the limitation of the independence assumption, and try to overcome it by attaching a small sequential autoregressive model to predict future tokens, after the main transformer block. This is again different from our approach, as we replace autoregressive sampling by auxiliary variables (see our Figure 2). This enables us to compute all tokens in parallel while leveraging the full transformer expressivity for each token. In comparison their autoregressive component (named ‘Sampler’) is less expressive. This claim is supported by their Figure 7. There, they find an additional speedup over multi-token prediction with independent tokens of less than 10% (3.4x vs 3.1x). We take full advantage of the token dependencies and thus observe a speedup of 100% (5.1x vs 2.5x) over independent prediction.
> Further fully complementary and compatible contributions include efficient finetuning using gated LoRA and efficient linear/quadratic decoding of future tokens.
>
> From a high-level perspective, our framework in Theorems 1 and 2 constitutes a dramatic change in the way prediction can be performed in language models: Instead of autoregressive sampling with multiple model calls, we enable parallel sampling in a single model call.
>
> Our auxiliary variables can be used as a plug-in replacement of [MASK] tokens used in multi-token prediction methods such as the two mentioned above. Intuitively, this provides strictly more information to the transformer ([MASK] tokens contain no additional information) and will result in a strictly higher speed-up when keeping the rest of these methods as they are proposed now.
>
>
> > Typically such multi-token prediction tasks are evaluated on Spec-Bench
>
> We appreciate this suggestion. Spec-Bench is an excellent resource, but training the required 7B-parameter chat models is beyond our current compute budget. We are exploring the possibility of additional resources and will report results if available.
>
> To partially compensate, we add a comparison to speculative decoding methods on the coding task (Section 3.2), showing that PTP achieves higher wall-clock speedups than an AR drafting model of the same size. The reviewer may find the comparison at this url: https://imgur.com/a/zlxbfxG
>
> > How do you guarantee the accuracy of the teacher model during PTP?
>
> We guarantee accuracy through the standard speculative-decoding error-correction step (Algorithm 1 in Appendix F), ensuring that the final output matches the teacher exactly.
>
> ***
>
> We again thank the reviewer for the feedback and look forward to the subsequent discussion.

---

> > ### Comment · Reviewer_Efeg · 2025-11-25
> >
> > Thank you for the clarifications.
> > This still doesn't answer 2 major questions. If error checking / correction is supposed to be done by SpecD doesn't it mean that the true speed up from this model is the reported metrics/2 ie number of correct tokens/2?
> > Regarding the speed up reported how does it compare with other MTP methods? EAGLE-2 is already close to 2.4x on average and in general it should be better in coding. Another question that arises is how do you run the model in inference mode? Is it fully optimized for inference (ie. compiled for inference using static graph, static memory allocations for tensors etc) because the improvement from Spec-D tends to be higher.
> > There needs to be some comparative baselines with existing methods and a story connecting the real impact to the problem statement.
> >
> > Overall, I think the contribution needs substantial modifications about the background work, original contributions and presentations of results. Therefore, I will keep my score.

---

> > > ### Author Response · Authors · 2025-12-04
> > >
> > > Thanks for your continued interest and feedback in our work! We understand that without a broader set of experiments, the reviewer was hesitant to recommend acceptance.
> > >
> > > ***
> > >
> > > > [Original review]: Typically such multi-token prediction tasks are evaluated on Spec-Bench
> > >
> > > > [From follow-up question] Regarding the speed up reported how does it compare with other MTP methods?
> > >
> > > Thanks again for the suggestion!
> > >
> > > We find that **PTP establishes a new state of the art on a variety of tasks** (Spec-Bench benchmark [3] at $T=0.7$) compared to a broad set of baselines:
> > >
> > > |                |   Multi-turn Conversation |   Translation |   Summarization |   Question Answering |   Mathematical Reasoning |   Retrieval-aug. Generation |   Average over tasks |
> > > |:---------------|:----------:|:-------------:|:---------------:|:----:|:----------------:|:-----:|:-----:|
> > > | **PTP (ours)** |       4.28 |      **3.35** |        **4.41** | **3.54** |     4.13 |  **4.27** |  **4.01** |
> > > | SAMD           |   **4.62** |          3.12 |            4.18 | 3.38 |             4.16 |  3.95 |  3.90 |
> > > | Eagle-2        |       4.54 |          3.19 |            3.81 | 3.41 |         **4.38** |  3.80 |  3.86 |
> > > | Hydra          |       3.90 |          2.88 |            2.95 | 3.21 |             3.90 |  3.37 |  3.37 |
> > > | Eagle          |       3.66 |          2.74 |            3.15 | 2.81 |             3.57 |  3.21 |  3.19 |
> > > | Medusa         |       2.70 |          2.18 |            2.12 | 2.24 |             2.67 |  2.26 |  2.36 |
> > > | SpS            |       2.12 |          1.39 |            2.16 | 1.80 |             1.92 |  2.26 |  1.94 |
> > > | Recycling      |       2.74 |          2.48 |            2.68 | 2.59 |             3.05 |  2.63 |  2.70 |
> > > | PLD            |       1.67 |          1.10 |            2.73 | 1.37 |             1.80 |  1.71 |  1.73 |
> > > | Lookahead      |       1.69 |          1.24 |            1.54 | 1.56 |             1.92 |  1.42 |  1.56 |
> > >
> > >
> > > This table compares the average number of accepted tokens, as this is directly proportional to the maximum speedup that can be obtained with efficient decoding [1, 2].
> > >
> > > [1] Li et al. “EAGLE-2: Faster Inference of Language Models with Dynamic Draft Trees” EMNLP 2025
> > >
> > > [2] Samragh et al. “Your LLM Knows the Future: Uncovering Its Multi-Token Prediction Potential” arXiv preprint arXiv:2507.11851 (2025)
> > >
> > > [3] Heming et al. "Unlocking Efficiency in Large Language Model Inference: A Comprehensive Survey of Speculative Decoding" ACL 2024
> > >
> > > > [From follow-up question] If error checking / correction is supposed to be done by SpecD doesn't it mean that the true speed up from this model is the reported metrics/2 ie number of correct tokens/2?
> > >
> > > That is partially correct. The speedup for any existing method is directly proportional to the average number of accepted tokens. It is true that the naive speedup is (accepted tokens)/2, but faster speeds can be achieved by with parallel verification, for example with quadratic decoding [2].
> > >
> > > [2] Samragh et al. “Your LLM Knows the Future: Uncovering Its Multi-Token Prediction Potential” arXiv preprint arXiv:2507.11851 (2025)
> > >
> > > > how do you run the model in inference mode?
> > >
> > > Our model can be optimized with the techniques you mentioned. Note that these tricks also apply to the teacher, so they don't necessarily result in a speedup relative to the teacher. They do of course result in an obverse wall clock speedup.
> > >
> > > ***
> > >
> > > We hope that the significant additional experiments and clarifications address all concerns of the reviewer.

---

### Official Review · Reviewer_kW1t · 2025-11-01

**Soundness:** 2
**Presentation:** 3
**Contribution:** 3
**Rating:** 6
**Confidence:** 4

**Summary:**

The paper proposes a method to improve **multi-token prediction (MTP)** by using $k$ auxiliary variables in the context which “represent” the next $k$ tokens. These auxiliary variables (sampled uniformly in $[0,1]$) contain information about the next generated tokens and are used to sample the $k$-th token ahead from the corresponding cumulative distribution function.

To train the model, the authors propose methods to retrieve these auxiliary random variables (e.g., from a teacher autoregressive model). The student model (the multi-token predictor) learns to reverse-engineer the process by predicting the same tokens as the teacher with the additional auxiliary variables. This way, training and inference match.

Experiments show improvements over standard MTP baselines.

**Strengths:**

- The approach is conceptually sound and addresses a relevant challenge: reducing autoregressive inference latency by enabling multiple token predictions in one forward pass.

- The method provides the model with additional structured information about future sampling decisions, potentially allowing more informed multi-token predictions than standard MTP approaches.

- Several design dimensions are explored, including loss functions and auxiliary-variable extraction strategies.

**Weaknesses:**

- **(Major) Role of auxiliary variables.** A stronger baseline would include the same number of additional “special” tokens as placeholders for future tokens (similar in spirit to the Pause-token approach [https://arxiv.org/abs/2310.02226]), trained to predict future tokens directly. Such a baseline would help clarify whether the auxiliary variables provide unique modeling benefits or primarily serve as additional learned positions.

- **(Minor) Training efficiency.** The highest-performing configuration relies on generating full sequences from a teacher model in order to extract auxiliary variables, which is computationally expensive.

- While the evaluation demonstrates gains on the selected setups, the method appears broadly applicable; including results on standard reasoning and coding benchmarks (e.g., TruthfulQA, HumanEval, GSM8K) would strengthen the empirical validation and illustrate generality.

Small typo: “which jointly predicts future tokens $t_k$” → “which jointly predicts future tokens $t_i, \ldots, t_k$”.

**Questions:**

How does performance change as the number of auxiliary variables (i.e., prediction horizon) increases?
Is there a point at which adding more auxiliary variables yields diminishing returns?

---

> ### Author Response · Authors · 2025-11-22
>
> We thank the reviewer for the thoughtful and constructive feedback. We will address the reviewer's comments point-by-point.
>
> ***
>
> > (Major) Role of auxiliary variables [; please compare to the] same number of additional “special tokens” as placeholders. [This] would clarify whether the auxiliary variables provide unique modeling benefits or primarily serve as additional learned positions
>
> Thank you for this suggestion! Using ‘special’ placeholder tokens indeed is a common way to train multi-token prediction models. The resulting tokens are only conditioned on the prompt tokens and thus independent of each other. We address this limitation in our ‘Independent prediction’ analysis in Section 2.4: Sampling tokens independently often produces invalid code (like `def numpy` or `import f`). As you suggest, our auxiliary variables can be used as a plug-in replacement of ‘special’ tokens. Intuitively, auxiliaries provide the transformer with strictly more information, resulting in strictly higher speed-up. Table 3 shows that using auxiliary variables increases the number of correct tokens per model call from 2.5 (independent) to 5.1. This confirms that the additional information is indeed helpful.
>
>
> By the suggestion of reviewers **[125r]** and **[6jCL]**, we also compare to speculative decoding in this figure https://imgur.com/a/zlxbfxG, which drafts tokens using a small AR model. PTP achieves additional speedups even at smaller model sizes.
>
> > (Minor) Training efficiency. The highest-performing configuration relies on generating full sequences from a teacher model in order to extract auxiliary variables, which is computationally expensive.
>
> Yes, we agree this could be costly, but in practice it is manageable. The dataset only needs to be generated once, before training begins. This is efficient because we optimize throughput rather than latency (as we want to generate a large dataset with different prompts). Note that we can also use a dataset or student predictions to generate training data, compare the last paragraph of section 2.2.1.
>
> > While the evaluation demonstrates gains on the selected setups, the method appears broadly applicable; including results on standard reasoning and coding benchmarks (e.g., TruthfulQA, HumanEval, GSM8K) would strengthen the empirical validation and illustrate generality.
>
> We agree; we believe PTP can generalize to reasoning and coding benchmarks. Running these tasks requires more compute than our current setup allows. We are exploring the possibility of additional resources and will report results if available. For now, we think that our comparisons to independent token prediction (section 3.2) and speculative decoding (see comparison mentioned in first answer) confirm the efficacy of our framework.
>
> > How does performance change as the number of auxiliary variables (i.e., prediction horizon) increases? Is there a point at which adding more auxiliary variables yields diminishing returns?
>
> Thank you for the suggestion! We ran this experiment and observed the following: The average number of correct tokens increases for longer prediction horizons, and we find diminishing returns:
>
> | Requested tokens | 1    | 2    | 4   | 8   | 12   | 16   |
> |:-----------------|-----:|-----:|----:|----:|-----:|-----:|
> | Correct tokens   | 0.91 | 1.60 | 2.6 | 4.3 |  4.9 |  5.1 |
>
> Smaller models reach this plateau sooner: In the scaling experiment (first answer), we find that smaller models achieve less correct tokens. In Table 7 in the appendix we find the same for the taxi dataset.
>
> ----
>
> We again thank the reviewer for the feedback and look forward to the subsequent discussion.

---

> > ### Comment · Reviewer_kW1t · 2025-11-26
> > **Response**
> >
> > I thank the authors for answering my questions and for running the additional experiment! It makes sense to observe some diminishing returns.
> >
> > As far as I understand, the independent prediction baseline is the standard MTP model of Gloeckle et al. (2024), where multiple heads decode future tokens from the final hidden states. I was asking something slightly different, i.e., adding some dummy tokens to the context and generating multiple future tokens in parallel. This baseline allocates more compute than the original MTP model, and it is also a little closer to the proposed PTP model.
> >
> > Overall, I keep my score and stay in favor of acceptance.

---

> > > ### Author Response · Authors · 2025-11-27
> > >
> > > Thanks for your helpful and encouraging answer! Let us clarify:
> > >
> > > The experiment suggested by the reviewer is exactly the one we do (Section 3.1 and 3.2). In detail: Both our PTP and the independent token prediction use the same architecture and have the same compute for each token. Our **auxiliary variables double the number of correct tokens per model call from 2.5 to 5.1** since they can coordinate token pairs (e.g. `import numpy` and `def fun`) unlike independent modeling (which also assigns high probability to unrealistic pairs like `import fun` and `def numpy`). We will make clearer that while we compare to the abstract modeling approach (independent prediction) of Gloeckle et al. (2024), we do not limit the model capacity for each token.
> > >
> > > We hope this addresses your remaining concern, and we are happy to address further questions!

---

> ### Author Response · Authors · 2025-12-04
>
> Thanks again for the positive review and for recommending acceptance of the paper. We would like to inform the reviewer about an additional experiment we performed due to their and reviewer Efeg's suggestion:
>
> ***
>
> > [...] the method appears broadly applicable; including results on standard reasoning and coding benchmarks (e.g., TruthfulQA, HumanEval, GSM8K) would strengthen the empirical validation and illustrate generality.
>
> Thanks again for the suggestion!
>
> We find that **PTP establishes a new state of the art on a variety of tasks** (Spec-Bench benchmark [3], which includes GSM8K, among others):
>
> |                |   Multi-turn Conversation |   Translation |   Summarization |   Question Answering |   Mathematical Reasoning |   Retrieval-aug. Generation |   Average over tasks |
> |:---------------|:----------:|:-------------:|:---------------:|:----:|:----------------:|:-----:|:-----:|
> | **PTP (ours)** |       4.28 |      **3.35** |        **4.41** | **3.54** |     4.13 |  **4.27** |  **4.01** |
> | SAMD           |   **4.62** |          3.12 |            4.18 | 3.38 |             4.16 |  3.95 |  3.90 |
> | Eagle-2        |       4.54 |          3.19 |            3.81 | 3.41 |         **4.38** |  3.80 |  3.86 |
> | Hydra          |       3.90 |          2.88 |            2.95 | 3.21 |             3.90 |  3.37 |  3.37 |
> | Eagle          |       3.66 |          2.74 |            3.15 | 2.81 |             3.57 |  3.21 |  3.19 |
> | Medusa         |       2.70 |          2.18 |            2.12 | 2.24 |             2.67 |  2.26 |  2.36 |
> | SpS            |       2.12 |          1.39 |            2.16 | 1.80 |             1.92 |  2.26 |  1.94 |
> | Recycling      |       2.74 |          2.48 |            2.68 | 2.59 |             3.05 |  2.63 |  2.70 |
> | PLD            |       1.67 |          1.10 |            2.73 | 1.37 |             1.80 |  1.71 |  1.73 |
> | Lookahead      |       1.69 |          1.24 |            1.54 | 1.56 |             1.92 |  1.42 |  1.56 |
>
>
> This table compares the average number of accepted tokens, as this is directly proportional to the maximum speedup that can be obtained with efficient decoding [1, 2].
>
> [1] Li et al. “EAGLE-2: Faster Inference of Language Models with Dynamic Draft Trees” EMNLP 2025
>
> [2] Samragh et al. “Your LLM Knows the Future: Uncovering Its Multi-Token Prediction Potential” arXiv preprint arXiv:2507.11851 (2025)
>
> [3] Heming et al. "Unlocking Efficiency in Large Language Model Inference: A Comprehensive Survey of Speculative Decoding" ACL 2024
>
> ***
>
> We hope this strengthens the recommendation for acceptance by the reviewer.

---

### Official Review · Reviewer_125r · 2025-11-02

**Soundness:** 2
**Presentation:** 2
**Contribution:** 2
**Rating:** 4
**Confidence:** 3

**Summary:**

Modern autoregressive models estimate the conditional probability of the next token given all the past tokens available in the context. One limitation of this class of models is that generating a sequence of $N$ tokens requires $N$ forward passes, hence resulting in possibly high latency. The paper proposes a solution to this problem by introducing a new approach, Parallel Token Prediction (PTP), enabling parallel generation of multiple tokens in parallel in a single forward pass. By incorporating auxiliary random variables into the model -- borrowing ideas from inverse autoregressive flows --  in PTP the generated tokens are not generated independently, in contrast to existing multi-token-prediction models. PTP models can be either trained from scratch or via distillation from standard autoregressive models. The experiments show some evidence that PTP models can effectively match the performance of their autoregressive counterparts yet significantly reducing latency.

**Strengths:**

- The paper presents a simple yet, to the best of the reviewer’s knowledge, novel approach in which the model learns to sample directly by conditioning on random variables, enabling it to anticipate and jointly predict future tokens.

**Weaknesses:**

- Paper clarity: the presentation of the paper can be improved in multiple points. See questions.
- The experimental evaluation is quite limited, focusing on a toy dataset and a small-scale distillation task. Given the potential of the idea, stronger empirical evidence would help validate its impact.
- While the provided arguments about the theoretical speedups guaranteed by PTP seem convincing, I believe the paper would greatly benefit from an empirical comparison between the latency of standard autoregressive models compared to the proposed frameworks.
- At multiple points in the paper (lines: 054-056, 441-442, 465-466) it is mentioned that the model proposed in [1] generates multiple independent tokens. I am not sure this is the case as in section 2.2 of [1] the authors state that "Different from Gloeckle et al. (2024), which parallelly predicts $D$ additional tokens using independent output heads, we sequentially predict additional tokens and keep the complete causal chain at each prediction depth". I would appreciate it if the authors can clarify this point.


[1] Liu, Aixin, et al. "Deepseek-v3 technical report." arXiv preprint arXiv:2412.19437 (2024).

**Questions:**

- In Eq. (6), shouldn't it be $P_{i+1} = P(t_{i+1}|t_{<i}, u_i)$?
- The l.h.s. of Eq. (7) does not seem correct. Rather, I think it should be written as $P(t_{k}|t_{<i}, u_i, ..., u_{k-1})$. Could the authors clarify?
- Could the authors provide intuition or evidence for potential performance gains relative to [1] (see weaknesses)?
- Have you explored how the performance of PTP evolves with scale, and how that compares to AR baselines?
- Can a PTP model trained to predict $N$ tokens be adapted at inference time to predict $M$ (with $M \neq N$) tokens?

[1] Liu, Aixin, et al. "Deepseek-v3 technical report." arXiv preprint arXiv:2412.19437 (2024).

---

> ### Author Response · Authors · 2025-11-22
>
> We thank the reviewer for the helpful and constructive feedback. Below is our point-to-point response to your comments:
>
> ***
>
> > In Eq. (6) [and (7)], shouldn’t it be $P_{i+1} = P(t_{i+1}|t_{<i}, u_i)$ [and $P(t_{k}|t_{<i}, u_i, ..., u_{k-1})$]? Could the authors clarify?
>
> Yes, thanks for catching that typo! As correctly noted, the distribution of the token at position $k$ only has access to tokens at positions $1, …, i-1$, and the auxiliaries at positions $i, …, k-1$.
>
> > At multiple points in the paper [...] it is mentioned that the model proposed in [1] generates multiple independent tokens, [but] the authors state that [they] "sequentially predict additional tokens and keep the complete causal chain at each prediction depth"
>
> You are correct, thank you for pointing this out. The reviewer is correct that the multi-token prediction in DeepSeek-v3 [1] does model dependencies through a small autoregressive model. We will update the manuscript to reflect this accurately.
>
> [1] Liu, Aixin, et al. "Deepseek-v3 technical report" arXiv preprint arXiv:2412.19437 (2024).
>
> > Could the authors provide intuition or evidence for potential performance gains relative to Deepseek-v3 [1]?
>
> Yes, our model offers more capacity for future tokens without a computational burden. In detail:
>
> Our approach generalizes multi-token prediction by allocating full-model compute that attends to every predicted future token and their dependencies, rather than relying on lightweight auxiliary heads. In our framework, a single full-transformer forward pass predicts all M future tokens in parallel, and each predicted token has its own full hidden state participating in attention, in all layers. In Deepseek-v3 [1], each future token is produced by a much smaller auxiliary autoregressive head with correspondingly limited compute. An ablation of the effect of adding a small auxiliary autoregressive component can be found in Figure 7 of [2], denoted there as ‘Sampler’. They find an additional speedup over multi-token prediction with independent heads of less than 10% (3.4x vs 3.1x). We observe a comparable additional speedup of our fully parallel model of over 100% (5.1x vs 2.5x). This can be attributed to limited capacity.
>
> Our additional flexibility comes with negligible computational cost. Computing attention over the potentially hundreds of tokens in the prompt dominates both compute and memory, so adding, for example, 16 extra predicted tokens contributes only a vanishing fraction of the total attention cost.
>
> [2]  Samragh et al. “Your LLM Knows the Future: Uncovering Its Multi-Token Prediction Potential” arXiv preprint arXiv:2507.11851 (2025)
>
>
> > Have you explored how the performance of PTP evolves with scale, and how that compares to AR baselines?
> > While the provided arguments about the theoretical speedups guaranteed by PTP seem convincing, I believe the paper would greatly benefit from an empirical comparison between the latency of standard autoregressive models compared to the proposed frameworks.
>
> We appreciate the suggestion and add this experiment for the rebuttal. Please see this figure at https://imgur.com/a/zlxbfxG. Two main findings:
>
> 1. PTP outperforms AR baselines of the same size. The reason is that for the same model size, an AR baseline can only produce one single sample per model call, but our PTP can produce several samples, of which several are correct.
> 2. Larger models produce more correct tokens per call, though their evaluation cost is higher, leading to a plateau in wall-clock speedup. Note that the bigger models have not fully converged yet, so we expect the plateau to eventually be slightly more in the favor of larger models.
>
> > Can a PTP model trained with $N$ tokens be adapted at inference time to predict $M$ [...] tokens?
>
> Yes, this works for $M<N$. Since our model is a causal transformer, we can feed fewer auxiliary variables to generate fewer tokens. As argued above, the computational impact of increasing $M$ to predict more tokens is usually negligible in comparison to the prompt tokens. For example, with our TinyLLama model on the coding task, we obtain:
>
> | Requested tokens |   1 |   2 |   4 |   8 |  12 |  16 |
> |:---|---:|----:|----:|----:|----:|----:|
> | Correct tokens (N=16)  | 0.9 | 1.6 | 2.6 | 4.3 | 4.9 | 5.1 |
>
> For $M>N$, additional training is recommended for best results. On the taxi dataset, we confirmed that finetuning for more tokens increases the average number of correct tokens. Supplementing the results already provided in Table 7 of the appendix of our paper, we found:
>
>
> | Requested tokens       |    4 |    8 |    64 |   100 |
> |:---|-----:|-----:|------:|------:|
> | Correct tokens (N=4)   | 3.95 | 7.21 | 15.00 | 15.49 |
> | Correct tokens (N=8)   | 3.91 | 7.55 | 20.66 | 21.40 |
> | Correct tokens (N=64)  | 3.89 | 7.50 | 31.82 | 38.15 |
> | Correct tokens (N=100) | 3.89 | 7.43 | 31.97 | 38.46 |
>
>
>
> ***
>
> We again thank the reviewer for the feedback and look forward to the subsequent discussion.

---

### Author Response · Authors · 2025-12-04

We thank the reviewers as well as the area chair for their time and effort. Let us summarize the contributions as well as the reviewer feedback below:

***

### Core contributions
- We propose parallel token prediction (PTP), a framework to predict several tokens in a single model call.
- Empirically, the model sets a new state of the art across diverse tasks, such as coding, question answering, RAG and others.

### Reviewers: Strengths

- Novel and interesting idea removing assumptions of previous works (all reviewers)
- Coding model: improved quality over independent prediction (all reviewers)

### Reviewers: Feedback

- Limited empirical validation; no real-world or large-scale benchmarks (125r, kW1t, Efeg, 5jCL).
    - **NEW**: We set a new state of the art on Spec-Bench across a variety of tasks (see below).
- No wall clock experiments; missing ablation against speculative decoding with different model sizes (125r, 5jCL)
    - **NEW**, see https://imgur.com/a/zlxbfxG: PTP outperforms speculative decoding by predicting several correct tokens per model call, also faster in wall clock time.
- Novelty concerns (Efeg):
    - **Clarification (approved by reviewer)**: BiTA uses independent prediction, Your LLM knows the future is autoregressive.


### New experiment: Spec-Bench

As suggested by reviewer Efeg, we finetune a Vicuna-7B chat model on ShareGPT completions to confirm that our PTP framework works across a variety of tasks.

We find that **PTP establishes a new state of the art on the benchmark**:

|                |   Multi-turn Conversation |   Translation |   Summarization |   Question Answering |   Mathematical Reasoning |   Retrieval-aug. Generation |   Average over tasks |
|:---------------|:----------:|:-------------:|:---------------:|:----:|:----------------:|:-----:|:-----:|
| **PTP (ours)** |       4.28 |      **3.35** |        **4.41** | **3.54** |     4.13 |  **4.27** |  **4.01** |
| SAMD           |   **4.62** |          3.12 |            4.18 | 3.38 |             4.16 |  3.95 |  3.90 |
| Eagle-2        |       4.54 |          3.19 |            3.81 | 3.41 |         **4.38** |  3.80 |  3.86 |
| Hydra          |       3.90 |          2.88 |            2.95 | 3.21 |             3.90 |  3.37 |  3.37 |
| Eagle          |       3.66 |          2.74 |            3.15 | 2.81 |             3.57 |  3.21 |  3.19 |
| Medusa         |       2.70 |          2.18 |            2.12 | 2.24 |             2.67 |  2.26 |  2.36 |
| SpS            |       2.12 |          1.39 |            2.16 | 1.80 |             1.92 |  2.26 |  1.94 |
| Recycling      |       2.74 |          2.48 |            2.68 | 2.59 |             3.05 |  2.63 |  2.70 |
| PLD            |       1.67 |          1.10 |            2.73 | 1.37 |             1.80 |  1.71 |  1.73 |
| Lookahead      |       1.69 |          1.24 |            1.54 | 1.56 |             1.92 |  1.42 |  1.56 |


This table compares the average number of accepted tokens, as this is directly proportional to the maximum speedup that can be obtained with efficient decoding [1, 2].

[1] Li et al. “EAGLE-2: Faster Inference of Language Models with Dynamic Draft Trees” EMNLP 2025

[2] Samragh et al. “Your LLM Knows the Future: Uncovering Its Multi-Token Prediction Potential” arXiv preprint arXiv:2507.11851 (2025)


### Outlook

Our parallel token prediction framework is a new paradigm to construct language models. It models token dependencies in a single model call, unlike autoregressive sampling and unlike discrete diffusion.

Our method is theoretically sound and sets a new state of the art in fast generation.

---

### Meta-Review · Area_Chair_xeEz · 2026-01-07

**Summary:**

This paper proposes a parallel token generation method for LMs. The core idea is to convert the sampling process of a categorical distribution to that of first sampling a continuous variable followed by a deterministic decoding. Obtaining a sequence of such continuous auxiliary variables allows one to generate multiple tokens at once with a deterministic function, which is the key aspect that sets the proposed work apart from existing multi token prediction works.

Two major concerns were raised: 1 lack of evaluations by Rev 125r and lack of novelty by Rev Efeg. The authors rebuttal provides significant more experimental results which the AC believes is sufficient. It also appears that Rev Efeg's critique on novelty is from a misunderstanding of the core idea. The AC believes that this a novel and interesting idea that is supported by (the newly added) results. The AC recommends accept.

**Reviewer Concerns:**

See summary. The AC believes that all major concerns are effectively addressed.

**Reviewer Scores:**

I would expect all reviewers willing to increase their ratings if they read and understood the paper and rebuttal.

---

### Decision · Program_Chairs · 2026-01-26

Accept (Poster)